# Regnase-1-mediated post-transcriptional regulation is essential for hematopoietic stem and progenitor cell homeostasis

Hiroyasu Kidoya [1], Fumitaka Muramatsu[1], Teppei Shimamura[2], Weizhen Jia[1], Takashi Satoh[3], Yumiko Hayashi[1], Hisamichi Naito[1], Yuya Kunisaki[4], Fumio Arai[4], Masahide Seki[5], Yutaka Suzuki[5], Tsuyoshi Osawa[6], Shizuo Akira[3,7] & Nobuyuki Takakura[1]

The balance between self-renewal and differentiation of hematopoietic stem and progenitor cells (HSPCs) maintains hematopoietic homeostasis, failure of which can lead to hematopoietic disorder. HSPC fate is controlled by signals from the bone marrow niche resulting in alteration of the stem cell transcription network. Regnase-1, a member of the CCCH zinc finger protein family possessing RNAse activity, mediates post-transcriptional regulatory activity through degradation of target mRNAs. The precise function of Regnase-1 has been explored in inflammation-related cytokine expression but its function in hematopoiesis has not been elucidated. Here, we show that Regnase-1 regulates self-renewal of HSPCs through modulating the stability of *Gata2* and *Tal1* mRNA. In addition, we found that dysfunction of Regnase-1 leads to the rapid onset of abnormal hematopoiesis. Thus, our data reveal that Regnase-1-mediated post-transcriptional regulation is required for HSPC maintenance and suggest that it represents a leukemia tumor suppressor.

[1] Department of Signal Transduction, Research Institute for Microbial Diseases, Osaka University, 3-1 Yamada-oka, Osaka, Suita, 565-0871, Japan. [2] Division of Systems Biology, Graduate School of Medicine, Nagoya University, 65 Tsurumai-cho, Nagoya, Showa-ku, 466-8550, Japan. [3] Department of Host Defense, Research Institute for Microbial Diseases, Osaka University, 3-1 Yamada-oka, Osaka, Suita 565-0871, Japan. [4] Department of Stem Cell Biology and Medicine/Cancer Stem Cell Research, Kyushu University, 3-1-1 Maidashi, Fukuoka, Higashi-ku 812-8582, Japan. [5] Department of Medical Genome Sciences Graduate School of Frontier Sciences, The University of Tokyo, Chiba 277-8561, Japan. [6] Division of Integrative Nutriomics and Oncology, Research Center for Advanced Science and Technology, The University of Tokyo, 4—6-1 Komaba, Tokyo, Meguro-ku 153-8904, Japan. [7] Laboratory of Host Defense, World Premier Institute Immunology Frontier Research Center, Osaka University, Osaka 565-0871, Japan. Correspondence and requests for materials should be addressed to H.K. (email: kidoya@biken.osaka-u.ac.jp) or to N.T. (email: ntakaku@biken.osaka-u.ac.jp)

The hematopoietic system is maintained over the lifetime of an organism through the well-orchestrated balance between self-renewal and differentiation of hematopoietic stem and progenitor cells (HSPCs)[1]. The HSPC compartment is heterogeneous and includes long-term hematopoietic stem cells (LT-HSCs) defined by their ability to give rise to all blood cell lineages and sustain life-long self-renewal. The vast majority of LT-HSCs is predominantly quiescent, remaining in the G0 phase of the cell cycle; the change to proliferative S+G2/M phase in response to hematological stress is a key event in hematopoietic homeostasis[2]. Quiescent LT-HSCs reside mainly in bone marrow (BM) niches, and their fate is controlled by multiple secreted and cell-surface molecules in the BM microenvironment[3]. Signals from the BM niche control HSPC fate via a variety of signaling pathways and transcription factors. Transcriptional regulation of gene expression through transcription networks plays crucial roles in hematopoiesis and in the maintenance of HSPCs[4]. Although various key transcription factors involved in HSPC homeostasis have been identified, regulatory mechanisms controlling the transcriptional network regulating hematopoiesis remain undetermined.

HSPCs maintain life-long hematopoiesis by self-renewal, which provides an opportunity for the accumulation of multiple genetic abnormalities. Accumulated chromosomal translocations and gene mutations can lead to malignant transformation of HSPCs and generation of leukemic stem cells (LSCs). It is widely accepted that LSCs acquire aberrant self-renewal capacity in contrast to normal HSPCs which have restricted self-renewal capacity and mostly remain in the quiescent state;[5] this results in the development of leukemia[6]. LSCs are also thought to be responsible for leukemia maintenance, therapy failure and disease relapse[7]. Acute myeloid leukemia (AML) is the most common type of leukemia in adults, characterized by the uncontrolled proliferation of abnormal and dysfunctional progenitor cells (blasts) in the BM. Transcriptional deregulation through aberrant expression and frequent mutation of transcription factors has been reported in AML patients[8]. Such abnormal transcriptional regulation leads to leukemogenesis and is crucially involved in the pathogenesis of AML.

The efficiency of mRNA translation is strictly controlled by post-transcriptional gene regulation. Cis-acting elements located in the 3′-untranslated region (3′UTR) of mRNA plays a key role in the modulation of mRNA stability[9,10]. These elements enable the recognition of target mRNA transcripts by RNA-binding proteins, and promote nuclease-dependent degradation[11,12]. The CCCH zinc finger protein Regnase-1 encoded by the ZC3H12A (MCPIP1) gene has been identified as a ribonuclease that suppresses gene expression through degradation of transcripts[13,14]. This protein acts as a negative regulator of inflammatory responses by destabilizing inflammation-related cytokines and transcription factor mRNAs such as interleukin-1β (IL-1β), IL-2, IL-6, IL-12p40, and c-Rel[13–16]. A recent study revealed that Regnase-1 binds to target mRNA via reorganization of the conserved stem-loop structure on the 3′UTR of these genes[17]. In addition to its immune response functions, Regnase-1 is also involved in a wide variety of biological processes such as brain development and adipogenesis by controlling cell differentiation and apoptosis[18,19].

In the present study, to elucidate cell-intrinsic mechanisms of HSPC homeostasis, we explored which molecules are essential for the regulation of gene expression. Using bioinformatics analysis of gene expression data during HSPC development, the ribonuclease Regnase-1 was identified as a major player. To study essential functions of Regnase-1 in HSPCs, we generated conditional knockout mice in which Regnase-1 was deleted specifically in HSPCs. Here, we document crucial roles of Regnase-1 for self-renewal of HSPCs. In addition, we show the relevance of the lack of Regnase-1 for leukemogenesis, associated with the aberrant expression of key regulators of hematopoiesis.

## Results

**Identification of genes important for HSPC maintenance.** To identify factors crucial for the determination of HSPC self-renewal and differentiation, we performed genome-wide gene expression analysis of Lineage⁻ Sca-1⁺ c-Kit⁺ (LSK) HSPCs in different developmental stages using available GEO data sets (GSE69760). The expression of multiple genes was different in HSPCs from adult BM and E14.5 FL. We screened genes encoding intercellular proteins for those highly expressed in adult HSPCs relative to HSPCs from embryos (Fig. 1a). Among these genes, we selected the gene encoding Regnase-1 because this molecule has been reported to associate with mesenchymal stem cell differentiation[20]. The amount of Regnase-1 expression in neonates was greater than in the fetus, and even greater in adults (Fig. 1b). To determine the expression profile of Regnase-1 in HSPC subpopulations, we isolated hematopoietic cells (HC; CD45⁺), LSK-HSPCs, immature and quiescent (CD34⁻ HSCs; CD34⁻ Flt3⁻ LSK), active (CD34⁺ HSCs; CD34⁺ Flt3⁻ LSK), and multipotent progenitors (MPPs; CD34⁺ Flt3⁺ LSK) from adult C57BL/6 WT mice[21–23]. The level of Regnase-1 mRNA was then determined by qRT-PCR. We found that Regnase-1 was relatively highly expressed in all HSPC subsets compared to the whole population of lineage-committed cells and differentiated progenitor cells (Fig. 1c, Supplementary Fig. 1a). Immunohistochemical staining of BM tissue from the femur revealed that Regnase-1 protein was predominantly present in c-Kit-positive cells including HSPCs (Fig. 1d).

**Effects of Regnase-1 deletion on BM HSPC populations.** To elucidate the function of Regnase-1 in HSPC development and/or maintenance, we analyzed mice harboring a floxed allele of the gene encoding Regnase-1[16] mated with mice expressing hematopoietic-specific Cre recombinase under the transcriptional control of vav promoter elements[24]. To confirm the deletion of Regnase-1 in CD34⁻ HSC, we performed qRT-PCR analysis of sorted CD34⁻ HSC from 8-week-old control Reg1^flox/flox (Regnase-1^flox/flox), Reg1^Δ/+ (Vav1-iCre; Regnase-1^flox/+), and Reg1^Δ/Δ (Vav1-iCre; Regnase-1^flox/flox) mice. As expected, Regnase-1 expression was almost completely absent in the CD34⁻ HSC of Reg1^Δ/Δ and reduced by approximately half in Reg1^Δ/+ (Fig. 1e). Reg1^Δ/Δ mice were born at predicted Mendelian ratios, but they exhibited growth reduction compared with their littermate controls. Flow cytometric analysis of HSPC subpopulations in adult BM revealed that Reg1^Δ/Δ mice possessed higher proportions of LSK-HSPCs and CD34⁻Flt3⁻ phenotypically defined CD34⁻ HSCs (Fig. 1f, g). Although the total BM cell count was decreased in Reg1^Δ/Δ mice relative to littermate controls, the absolute numbers of LSK-HSPCs were significantly increased (Fig. 1h). Furthermore, a marked increment in the absolute number of CD34⁻ HSCs and CD34⁺ HSC in Reg1^Δ/Δ mice was observed (Fig. 1i). Consistent with these results from flow cytometry, histological analysis revealed a significant expansion of Sca1⁺ c-Kit⁺ HSPCs in femur sections from Regnase-1-deficient animals (Fig. 1j). In summary, our findings suggest that Regnase-1 is widely expressed in HSPCs and contributes to regulation of CD34⁻ HSC proliferation and differentiation.

**Loss of Regnase-1 leads to accumulation of immature HSPCs.** Definitive hematopoiesis starts in the aorta-gonad-mesonephros (AGM); HSC translocate to the fetal liver (FL) where they undergo expansion and differentiation. Unlike the FL, molecular

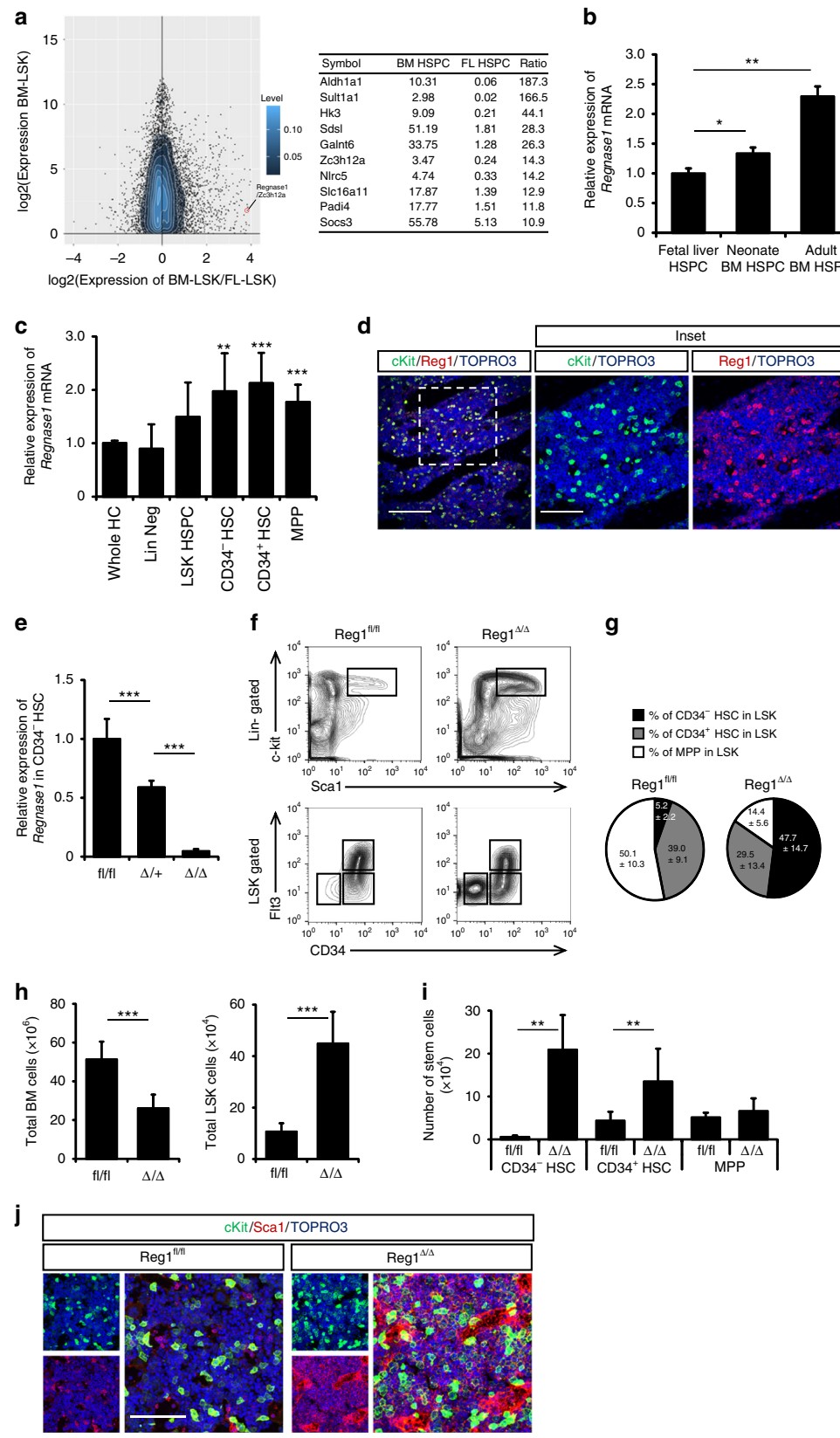

cues from the adult BM microenvironment induce growth arrest of HSCs in the G0 phase of the cell cycle and maintain HSC quiescence[25]. Because Regnase-1 expression is upregulated in HSPCs of adult BM compared with those of the fetus, this suggests the involvement of Regnase-1 in the regulation of the HSC

pool. Therefore, we investigated when the anomalous increase of HSPCs occurs in Regnase-1-deficient mice. Flow cytometric analysis of FL cells from E15.5 embryos and BM cells from 1-day-old neonates revealed no differences in the numbers or proportions of HSPCs. However, we found abnormal accumulations of

**Fig. 1** Regnase-1 is expressed in HSPCs and is involved in maintenance of the HSC pool. **a** Mean difference plot of mRNA expression in Lineage⁻ Sca-1⁺ c-Kit⁺ (LSK) HSPCs from adult BM and HSPCs from E14.5 FL using the GEO data sets (GSE69760). Genes with expression levels increased ≥10-fold in adult BM and encoding intra-cellar proteins are listed in the table on the right. **b** Quantitative RT-PCR of *Regnase-1* expression in isolated LSK cells from E14.5 fetal liver, neonate BM, and adult BM ($n = 3$ per group). Data are expressed as fold-change relative to fetal liver HSPCs. **c** *Regnase-1* mRNA expression in lineage-negative cells (Lin Neg), Lin⁻ cKit⁺ Sca-1⁺ cells (LSK HSC), LSK CD34⁻ Flt3⁻ (CD34⁻ HSC), LSK CD34⁺ Flt3⁻ (CD34⁺ HSC), and LSK CD34⁺ Flt3⁺ (MPP) prepared from BM of adult mice (8-week-old) ($n = 3$). **d** Immunohistochemical staining of Regnase-1 in BM. The right-hand panels show a higher magnification of the areas indicated by the boxes. The scale bars represent 200 μm and 100 μm (insets). **e** Loss of Regnase-1 expression in CD34⁻ HSCs was confirmed by Quantitative RT-PCR of adult Vav1-iCre; Regnase-1ᶠˡᵒˣ/ᶠˡᵒˣ (Δ/Δ), Vav1-iCre; Regnase-1ᶠˡᵒˣ/⁺ (Δ/+) and control Regnase-1ᶠˡᵒˣ/ᶠˡᵒˣ (fl/fl). The level of Regnase-1 mRNA of the fl/fl mice was set at 1.0 ($n = 3$ mice of each genotype). **f–i** Representative flow cytometric analysis (**f**), quantitative and statistical analyses of HSPCs populations (**g**), and the numbers (**h, i**) in the BM of 8-week-old control (fl/fl) and Regnase-1-KO (Δ/Δ) mice ($n = 10$ per group). **j** Immunohistochemical staining of Sca-1⁺ c-Kit⁺ HSPCs in control and Regnase-1-KO (Δ/Δ) BM. The scale bars represent 100 μm. Error bars indicate mean ± SD. *$p < 0.05$; **$p < 0.01$, ***$p < 0.005$, Tukey–Kramer multiple comparison test in **b**, **c**, *t*-test in **c**, **h**, **i**

LSK-HSPCs and CD34⁻ HSCs in 2-week-old Reg1^{Δ/Δ} mice (Fig. 2a, b). This suggests an inhibitory role for Regnase-1 in HSPC proliferation and that it could contribute to the maintenance of CD34⁻ HSC quiescence. We investigated the activity of Regnase-1-deficient CD34⁻ HSC by analyzing the cell surface markers CD150 and CD48, useful for determining HSC quiescence;[26] quiescent HSC subsets are defined as the CD150⁺ CD48⁻ LSK population. We determined the frequency and number of quiescent HSCs in Regnase-1-deficient mice, but found no obvious changes compared with HSCs from WT mice (Reg1^{fl/fl}). On the other hand, the frequencies and numbers of CD150⁺CD48⁺ LSK-HSPCs and CD150⁻CD48⁺ LSK-HSPCs were significantly increased, and these cells were actively cycling (Fig. 2c, d, Supplementary Fig. 1b, c). Generally, CD150⁺ CD48⁺ LSK-HSPCs are defined as differentiating HSPCs[26]. However, CD48 expression on HSCs was also reported as a marker for cell cycle activation[2,27,28], and CD150⁺CD48⁺LSK cells include HSCs with colony-forming ability and long-term multi-lineage reconstitution potential similar to LSK-SLAM cells[29,30]. Moreover, the increased CD150⁺ CD48⁺ LSK population in Reg1^{Δ/Δ} mice did not express the CD34 marker of differentiated HSC, suggesting that Regnase-1 deficiency resulted in increase of activated CD34⁻ HSCs in BM.

**CD34⁻ HSC function is impaired by deletion of Regnase-1**. We next examined whether increased LSK-HSPCs and CD34⁻ HSCs in Reg1^{Δ/Δ} retain the ability to undergo normal hematopoietic differentiation in an in vitro CFC assay. LSK-HSPCs from Reg1^{Δ/Δ} mice exhibited no defects in colony formation and frequencies of different types of colonies compared with their littermate controls (Fig. 2e). In contrast, CD34⁻ HSCs from Reg1^{Δ/Δ} mice demonstrated an approximately 2-fold decrease in colony number with slight effect on frequencies of colony types (Fig. 2e). These data indicate that Regnase-1 is involved in the differentiation of CD34⁻ HSCs. To test the quality of Regnase-1-deficient CD34⁻ HSCs in vivo, we performed competitive BM transplantation (BMT) assays. Sorted CD34⁻ Flt3- LSKs from the BM of Reg1^{Δ/Δ} or control mice (CD45.2⁺) mixed with congenic (CD45.1⁺) WT BM cells were transferred into lethally (10 Gy)-irradiated WT (CD45.1⁺) mice. Recipient BM and peripheral blood (PB) cells were then analyzed by flow cytometry 16 weeks thereafter. The number of Reg1^{Δ/Δ}-deficient donor–derived LSK-HSPCs was 3.4-fold lower than in control BM, but the number of CD34⁻ HSCs from Reg1^{Δ/Δ}-deficient donors was 3.2-fold higher than in controls (Fig. 2f, g). These data suggest a diminished contribution of Reg1^{Δ/Δ} CD34⁻ HSCs to CD34⁺ HSCs and MMPs relative to control cells under competitive conditions, at the same time as the proliferative activity of Reg1^{Δ/Δ} CD34⁻ HSCs is increased.

Next, to investigate the long-term repopulating potential of Reg1^{Δ/Δ} CD34⁻ HSCs, we analyzed donor cell reconstitution in the peripheral B lymphoid (B220), T lymphoid (CD3), and myeloid (CD11b) populations of recipient mice. Although Reg1^{Δ/Δ} animals retained the ability to produce both myeloid and lymphoid lineages, Reg1^{Δ/Δ} CD34⁻ HSCs contributed to only 16.3% of B cells, 20.5% of T cells, and 10.5% of myeloid cells. This was a markedly lower contribution compared with dominant effects of HC lineage cells from control CD34⁻ HSCs (Fig. 2h, i). By analyzing earlier time points one month after transplantation, we found increased CD34⁺ LSK cells, MEPs and GMPs in the group transplanted with CD34⁻ HSCs from Regnase-1-deficient mice (Supplementary Fig. 1d). This suggests that Regnase-1 deficiency results in abnormal regulation of cell growth and induces excessive proliferation.

Together, these results indicate that Regnase-1-deficient CD34⁻ HSCs engraft normally, but their repopulation potential is significantly attenuated relative to control, consistent with the results from the in vitro colony forming assays. In summary, these data suggest that Regnase-1 plays an important role in regulating the balance between proliferation and differentiation of CD34⁻ HSCs.

**Regnase-1 regulates HSPC cell cycle activity**. Results from Regnase-1-deficient mice suggest that the balance between quiescence and proliferative status of CD34⁻ HSC is regulated by this enzyme. To identify the genes differentially expressed between control and Reg1^{Δ/Δ} CD34⁻ HSCs, global gene expression profiling was performed. Gene ontology (GO) terms related to cell cycle regulation and hematopoietic differentiation were enriched in Reg1^{Δ/Δ} CD34⁻ HSCs (Fig. 2j). Based on this result, we assessed whether the activity of Regnase-1 in HSPC maintenance was by means of regulating the cell cycle and/or apoptosis. To evaluate the cell cycle status of CD34⁻ HSCs and CD34⁺ HSC from Reg1^{Δ/Δ} mice, cells were labeled with EdU and 7-AAAD and analyzed for G0/G1, G2, and G2/M populations. As expected, Regnase-1 deletion led to an increase in the number of cells in all cell cycle phases in both the CD34⁺ HSC and CD34⁻ HSC fractions, and an especially notable increase in cell number in S phase CD34⁻ HSCs (Fig. 3a, b). In contrast, there was only a slight effect on the cell cycle of MPPs and lineage-committed cells (Supplementary Fig. 1e). These results indicate that deletion of Regnase-1 in HSPCs mainly affects cycling of CD34⁻ HSCs and CD34⁺ HSCs. Consistent with this, immunohistochemistry confirmed that BrdU incorporation was also increased in c-Kit-positive HSPCs from Reg1^{Δ/Δ} BM (Fig. 3c, d). Next, we performed Pyronin Y/Hoechst 33342 staining to distinguish between cells in the quiescent G0 phase and the proliferating G1-phase. Flow cytometric analysis revealed a marked increase in the number of cells at all phases of the cell cycle, but the proportion of proliferating cells in G1 and S/G2/M (Pyronin Y⁺ Hoechst⁻ and Pyronin Y⁺ Hoechst⁺) was increased relative to quiescent cells at G0 (Pyronin Y⁻ Hoechst⁻) in CD34⁺ HSCs from

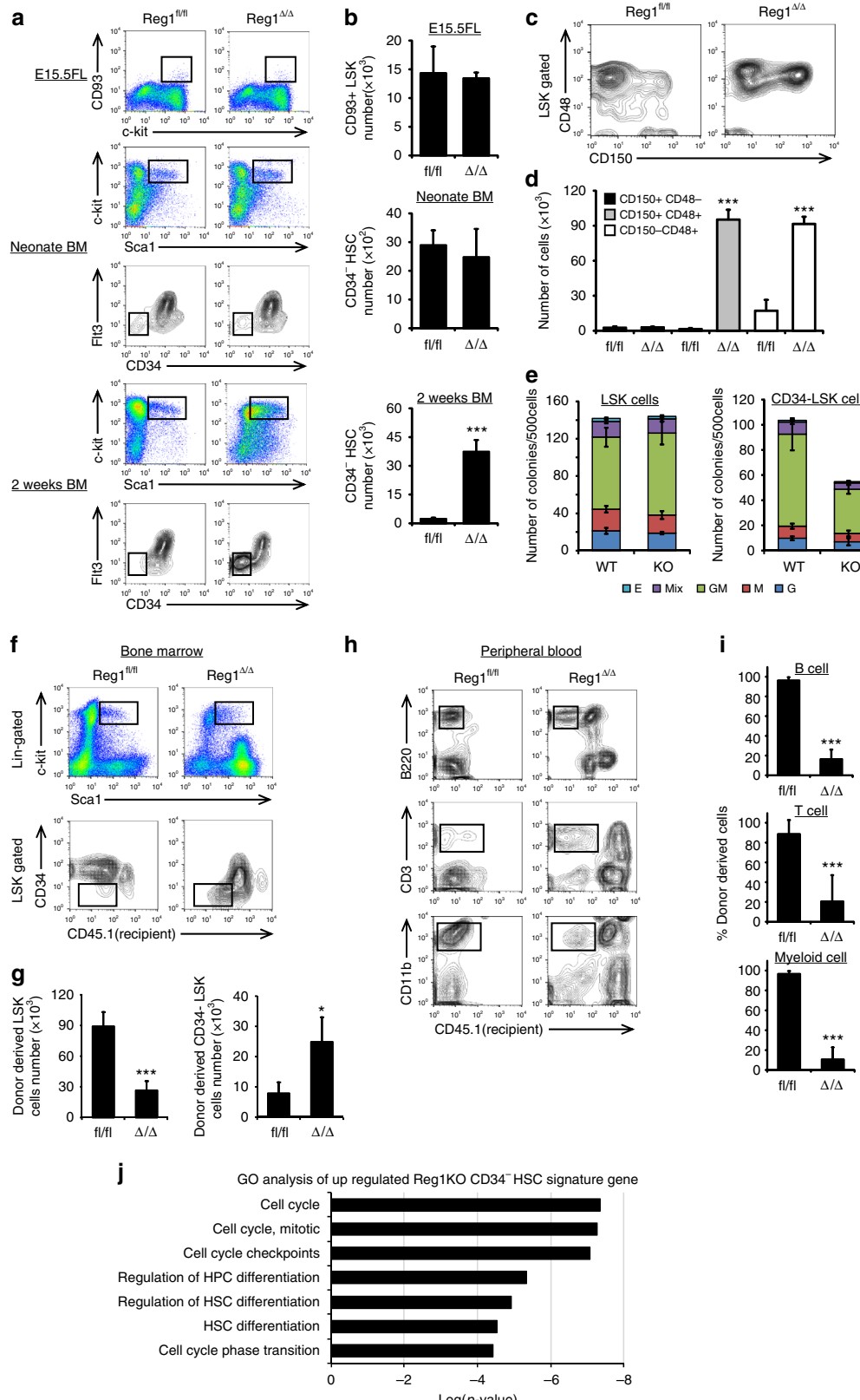

Reg1$^{\Delta/\Delta}$ BM (Fig. 3e, f). Moreover, we investigated the expression of the quiescence-associated genes p21/Cip, P27/Kip1 and p57/Kip2 by qRT-PCR. Down-regulation of p21/Cip and p57/Kip2 was observed in sorted Reg1$^{\Delta/\Delta}$ CD34$^-$ HSCs, more than in control CD34$^-$ HSCs (Fig. 3g).

Next, we assessed apoptosis using flow cytometry with Annexin V and 7-AAD staining. The ratio of the apoptotic fractions in the CD34$^-$ HSC and CD34$^+$ HSC compartments was no different in cells from Reg1$^{\Delta/\Delta}$ or control BM (Fig. 3h, i). Therefore, we concluded that the increase of HSPCs in Reg1$^{\Delta/\Delta}$

**Fig. 2** Regnase-1 deficiency leads to expansion of immature HSPCs. **a** Representative flow cytometric analysis of HSPC populations in the FL of E15.5 embryo, BM of 1-day-old neonate, BM of 2-week-old control (fl/fl) or Regnase-1-KO ($\Delta/\Delta$) mice ($n = 3$ per group). **b** The numbers of LSK-HSCs or CD34$^-$ HSCs in the total FL or BM cells assessed by flow cytometry. **c** Representative flow cytometric analysis of LSK cells for CD48 and CD150 expression of 8-week-old control (fl/fl) or Regnase-1-KO ($\Delta/\Delta$) ($n = 3$ per group). **d** Percentage of CD150$^+$ CD48$^-$ cells, CD150$^+$ CD48$^+$ cells, and CD150$^-$ CD48$^+$ cells of control (fl/fl) and Regnase-1-KO ($\Delta/\Delta$) mice. **e** Colony-forming ability of LSK-HSCs and CD34$^-$ HSC from control (fl/fl) or Regnase-1-KO ($\Delta/\Delta$) mice BM. G CFU-granulocyte; GM CFU-granulocyte/monocyte; M CFU-monocyte; E CFU-erythroid; and mix mixed CFU-granulocytes, monocyte, erythroid, and megakaryocyte. Data show the means ± SD ($n = 5$; 3 independent experiments). **f** Competitive repopulation assay of lethally irradiated (10 Gy) WT recipients (CD45.1) transplanted with BM of either control (fl/fl) or Regnase-1-KO ($\Delta/\Delta$) mice. Chimerism was analyzed in recipient mice 16 weeks after injection of $1 \times 10^3$ of CD34$^-$ HSCs from control or Regnase-1-KO mice (CD45.2) with $5 \times 10^5$ competitor BM cells (CD45.1). Donor-derived cell ratios (CD45.1$^-$) in BM HSPC and CD34$^-$ HSC populations were determined by flow cytometry ($n = 5$ per group). **g** The number of LSK or CD34$^-$ LSK cells in the BM of mice 16 weeks after transplantation. **h** Hematopoietic engraftment of transplanted HSPCs for reconstitution of peripheral blood was analyzed by flow cytometry at 16 weeks after transplantation ($n = 5$ per group). **i** The contribution of transplanted donor-derived cells (CD45.1$^-$) to B-cells (B220$^+$), T-cells (CD3$^+$), and myeloid cells (CD11b$^+$) from control (fl/fl) or Regnase-1-KO ($\Delta/\Delta$) HSPCs are shown. Data summated from 3 independent experiments and an average of 6 recipient mice in each group with SD. **j** Gene ontology (GO) analysis of genes upregulated in the CD34$^-$HSCs from Regnase-1-KO BM based on the GO biological process annotations. Error bars indicate mean ± SD. *$p < 0.05$; **$p < 0.01$, ***$p < 0.005$, two-sided $t$-test in **b**, **d**, **g**, **i**

BM was due to cell cycle progression, and that apoptosis was not responsible.

Next, to elucidate the function of Regnase-1 following hematopoietic stress, we subjected Reg1$^{\Delta/\Delta}$ mice to myelotoxic stress using 5-FU. Reg1$^{\Delta/\Delta}$ mice were more sensitive to repetitive 5-FU administration, and died significantly earlier than control Reg1$^{flox/flox}$ mice (Fig. 3j). To evaluate the impact of Regnase-1 deletion during BM ablation and recovery, the kinetics of HSC repopulation were established after a single injection of 5-FU. Consistent with results under the steady state, LSK-HSCs from Reg1$^{\Delta/\Delta}$ mouse BM rapidly increased more than in control LSK-HSCs after 5-FU treatment. The absolute numbers of LSK-HSCs on day 4 after 5-FU treatment also significantly increased in the BM of Reg1$^{\Delta/\Delta}$ mice relative to controls (Fig. 3k, l). 5-FU induces apoptosis of proliferating cells, but many quiescent G0 phase LSK-HSCs in Regnase-1-deficient mice show resistance to 5-FU and may lead to subsequent increase. After 5-FU treatment, Regnase-1 expression in LSK-HSCs from WT mice was up-regulated, coinciding with HSPC expansion (Fig. 3m). These data demonstrate that Regnase-1 has a critical role in the suppression of cell cycle entry and thereby, Reg1 deficiency induces excessive activation and proliferation of HSPCs after myeloablative stress.

**Regnase-1 deficiency causes impaired hematopoietic function.** Because Regnase-1 deficiency results in an excessive expansion of HSPCs and a decline in the maintenance of the hematopoietic system, we investigated whether Regnase-1 deletion leads to overt disease. We followed the long-term survival and changes in hematopoiesis of Reg1$^{\Delta/\Delta}$ mice. Strikingly, all such mice gradually developed abnormal hematopoiesis, and lost weight, and died within 110 days (Fig. 4a). Macroscopically, 10-week-old Reg1$^{\Delta/\Delta}$ mice exhibited severe splenomegaly and mesenteric lymphade-nopathy (Fig. 4b, c). Marked decreases of red blood cells, platelets, and hemoglobin concentration, and slight or no alteration of erythropoiesis and megakaryopoiesis were observed in Reg1$^{\Delta/\Delta}$ mice (Supplementary Fig. 2a–e). PB of these 8 week-old mice had an elevated granulocyte population (Gr1$^+$) and reduced T lymphoid population (CD3$^+$) relative to control Reg1$^{flox/flox}$ animals (Fig. 4d, e). Reg1$^{\Delta/\Delta}$ BM also contained significantly expanded Gr1$^+$ and Mac1$^+$ myeloid populations and reduced lymphoid populations (Fig. 4f, g). Juvenile Reg1$^{\Delta/\Delta}$ mice (2-weeks-old) also exhibited moderate abnormalities in lineage-committed cells in the PB and BM (Supplementary Fig. 3a, b). On the other hand, hematopoietic-specific Regnase-1 hetero-knockout Reg1$^{\Delta/+}$ mice (Vav1-iCre; Regnase-1$^{flox/+}$) were viable, and showed normal growth and appearance until they were 6-months-old. They then showed weakness later, and almost all died within 14 months (Fig. 4h). Reg1$^{\Delta/+}$ mice also displayed severe splenomegaly and

mesenteric lymphadenopathy (Fig. 4i, j), and expansion of LSK-HSPC and CD34$^-$ HSC subpopulations in BM (Fig. 4k, l). Flow cytometric analysis of PB and BM cells of the Reg1$^{\Delta/+}$ mice revealed a tendency for increased myeloid populations and decreased lymphoid populations, similar to Reg1$^{\Delta/\Delta}$ mice (Fig. 4m). This suggests that a single Regnase-1 allele is insufficient for maintaining normal hematopoiesis. Collectively, these results imply an indispensable role for Regnase-1 in the maintenance of normal hematopoiesis and that a reduction of Regnase-1 expression in HSPCs leads to the development of pathological hematopoiesis.

**Loss of Regnase-1 leads to aberrant differentiation.** We next tested whether loss of Regnase-1 in HSPCs relates to the development of leukemia. Aberrant blast cells with nucleoli were observed in the PB of 8-week-old Reg1$^{\Delta/\Delta}$ mice and were also seen in the BM (Fig. 5a, b, Supplementary Fig. 4a, b). Mono-nuclear cells in PB and BM were analyzed further to reveal the variations in disease phenotype. In PB from Reg1$^{\Delta/\Delta}$ mice, we observed 1–14% of lymphoblasts, 16–22% of atypical lymphocytes exhibiting abnormal nuclear morphology, 2–4% of mature myelocytes and metamyelocytes, and 1–16% of atypical cells including stab cells in the neutrophil population. In addition, 3–8% of monoblasts and 6–20% of atypical mononuclear cells were seen, and proerythroblastic-polychromatophilic erythroblasts were also observed. Furthermore, 1% of myeloblasts was present, and total white blood cell counts were increased; therefore we diagnosed abnormal hematopoiesis, such as MPN (Fig. 5c, d). Reg1$^{\Delta/+}$ mice also had a similar phenotype (Supplementary Fig. 4c). BM of Reg1$^{\Delta/\Delta}$ mice manifested no significant differences in frequencies of neutrophil populations, but had an expanded monocyte and reduced lymphocyte population. In addition, morphologically abnormal cell populations with increased cell volume or irregular nuclear shape were present in the leukocyte population i.e., neutrophils, monocytes, and lymphocytes, in Reg1$^{\Delta/\Delta}$ mouse BM (Supplementary Fig. 4d). Consistent with abnormalities in the mononuclear cells, LSK HSPCs from Reg1$^{\Delta/\Delta}$ mice displayed enlarged nuclei and aberrant cell shapes (Fig. 5e). These results suggest that Regnase-1 is required to maintain normal hematopoiesis and that disrupted expression of Regnase-1 leads to aberrant differentiation. In recipients transplanted with MNCs from BM of Reg1$^{\Delta/\Delta}$ mice, abnormal proliferation of donor-derived HSPCs, splenomegaly, and an abnormal hematopoietic phenotype of PB was observed, but blasts were not present in the PB. These results indicate that Reg1$^{\Delta/\Delta}$ mouse-derived aberrant cells are transplantable (Supplementary Fig. 5a–d). The phenotype of Regnase-1-deficient

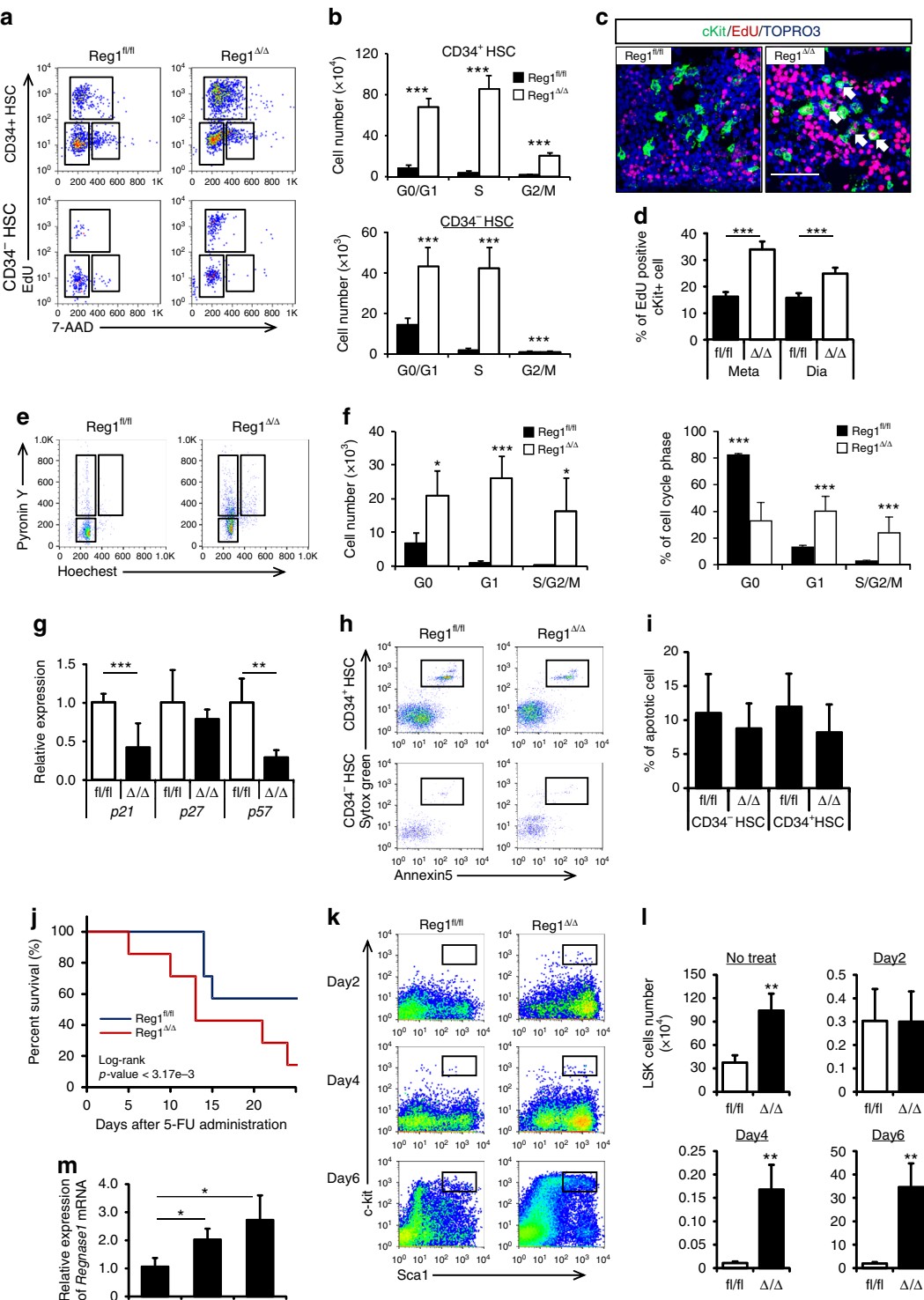

mice is not completely consistent with AML pathology, but does exhibit aberrant differentiation and proliferation of the HC.

To confirm the relevance of Regnase-1 deficiency for AML pathogenesis, we undertook RNA-seq-based genome-wide gene expression profiling using CD34− HSCs collected from control Reg1$^{flox/flox}$ and Reg1$^{\Delta/\Delta}$ mice BM. RNA-seq revealed that the expression of many genes was altered in Reg1$^{\Delta/\Delta}$ mice (Fig. 5f). Consistent with the morphological analysis, gene set enrichment analysis (GSEA) revealed that the genes downregulated in CD34− HSC following Regnase-1 deletion were statistically significantly

similar to an *a priori*-defined set of genes in HSCs from Pu.1-deficient mice, taken to model AML (Fig. 5g). A comparison of gene expression sets between CD34− HSCs from Regnase-1-deleted mice and BM from human AML-M1, M2, and M3 patients indicated significantly similar concordant differences in gene expression between Regnase-1-deleted CD34− HSCs and human AML cells (Fig. 5h). Although *Regnase-1* mRNA expression was not associated with prognosis, patients whose AML cells exhibited gene expression profiles similar to Regnase-1-deficient CD34− HSCs had a poor prognosis (Fig. 5i,

**Fig. 3** Loss of Regnase-1 expression results in accelerated CD34− HSC cell cycle progression. **a** Cell-cycle analysis of control (fl/fl) or Regnase-1-KO (Δ/Δ) BM CD34− HSC and CD34+ HSC populations using EdU/7AAD staining. Dot plots indicating the frequency of HSPCs in G0/G1 (EdU− 7AAD−), G2/M (EdU+), and S (EdU+ 7AAD+) phase of cell cycle (n = 3 per group; 3 independent experiments). **b** Quantification of HSPCs as shown in **a**. **c** Detection of proliferative cells in BM section of control (fl/fl) or Regnase-1-KO (Δ/Δ) mice by EdU labeling. Representative image showed EdU staining together with c-Kit and Nuclei (TOPRO3) counter-staining. Scale bars represent 50 μm. **d** Quantification of the ratio of EdU-positive proliferating cells in c-Kit-positive cells. **e** Representative cell cycle plots of CD34− HSC populations in BM from control (fl/fl) or Regnase-1-KO (Δ/Δ) mice using Hoechst 33342/pyronin Y staining (n = 3 per group; 3 independent experiments). **f** Representative data showing number and ratio of cells in the G0, G1, and S/G2/M phases as shown in **e**. **g** qRT-PCR analysis of cell cycle regulatory genes in CD34− HSCs from control (fl/fl) and Regnase-1-KO (Δ/Δ) mice (n = 3 per group). **h** Flow cytometric analysis of apoptotic populations in CD34− HSCs and CD34+ HSCs of control (fl/fl) or Regnase-1-KO (Δ/Δ) mice BM with a combination of annexin V and SYTO Green staining (n = 3 per group; 3 independent experiments). **i** Ratio of apoptotic cells (annexin V+ SYTO Green+). Data are the means ± SD. **j** Kaplan-Meier survival curve of control (fl/fl) and Regnase-1-KO (Δ/Δ) mice after weekly administration of 5-FU (150 mg/kg) (n = 7). P values were calculated by the logrank test. **k** Representative dot plot depicting frequency of phenotypically defined HSPCs at the indicated days following a single injection of 5-FU. **l** Absolute LSK cell numbers from control (fl/fl) and Regnase-1-KO (Δ/Δ) mice at the indicated days after 5-FU treatment (n = 3). **m** Expression Regnase-1 mRNA in LSK cells of control (fl/fl) mice at the indicated days after 5-FU treatment (n = 3). Error bars indicate mean ± SD. *p < 0.05; **p < 0.01, ***p < 0.005, two-sided t-test in **b**, **d**, **f**, **g**, **i**, **m**, **l**

Supplementary Data 1). In addition, we found that overexpression of Regnase-1 suppresses the exuberant growth of THP1 and HL60 cells (Fig. 5j).

**Regnase-1 regulates the stability of Gata2 and Tal1 mRNA.** Regnase-1 has been reported to control target gene expression by recognizing the specific 3′UTR elements of mRNA and to facilitate ribonuclease-mediated degradation[31]. To identify Regnase-1 target mRNAs in CD34− HSCs, we constructed luciferase reporter vectors by inserting the 3′UTRs of candidate mRNAs into the 3′ site of the luciferase expression cassette and then determining whether these 3′UTRs were actually Regnase-1 binding sites for degradation. Because Regnase-1 degrades various target mRNAs[32] and RNA-Seq analysis showed that the expression of numerous genes is affected by Regnase-1 deficiency (Fig. 5f), we focused on transcription factors that affect the expression of a wide range of genes important for hematopoiesis, such as Gata2, Tal1, Bmi1, Ctnnb1, Runx1, Hoxb4, and Egr1. In accordance with a previous report[33], co-transfection of luciferase reporter vectors of candidate genes together with the Regnase-1 expression vector resulted in suppression of luciferase activity caused by the RNase-activity of Regnase-1, relative to co-transfection with Regnase-1 lacking the RNase activity vector (Fig. 6a, Supplementary Fig. 6). Next, the identity of the candidate genes affected by Regnase-1 overexpression was confirmed by real-time QPCR. We concluded that Gata2 and Tal1 were prime targets of Regnase-1, because expression of these mRNAs was significantly upregulated in CD34− HSCs from Reg1Δ/Δ mice (Fig. 6b). To confirm that the degradation of Gata2 and Tal1 mRNA is directly induced by Regnase-1, we performed Northern blotting. Overexpressing Regnase-1 in THP1 cells attenuated Gata2 and Tal1 mRNA expression by accelerating the degradation of these mRNAs, but did not alter CD34 mRNA degradation. (Fig. 6c, Supplementary Fig. 7a). Moreover, the half-lives of Gata2 and Tal1 mRNA were extended in Regnase-1-deficient CD34− HSCs relative to CD34− HSCs from control mice (Fig. 6d). In summary, our findings suggest that Gata2 and Tal1 identified by bioinformatics analysis are indeed recognized and degraded by Regnase-1.

**Regnase-1 regulates degradation of Gata2 and Tal1 mRNA.** Next, we investigated whether inhibition of Gata2 and Tal1 in CD34− HSCs of Reg1Δ/Δ mice rescues the excessive HSPC self-renewal and subsequent development of aberrant differentiation. We did this by infection with an shRNA virus and confirmed knock-down of the expression of the targeted gene (Fig. 7a). To investigate the effect of Gata2 and Tal1 knock-down on the

proliferation and differentiation of Reg1Δ/Δ CD34− HSCs, we performed colony-forming assays. As expected, shRNA-mediated suppression of either or both Gata2 and Tal1 expression partially rescued the colony-forming ability of CD34− HSCs from Reg1Δ/Δ mice, whereas scrambled control shRNA had no effect (Fig. 7b). However, knock-down of Bmi1 (non-candidate control gene) did not alter the colony forming ability of Reg1Δ/Δ CD34− HSCs, suggesting that aberrant proliferation of these CD34− HSCs is induced by excessive expression of Gata2 and Tal1. In addition, we investigated the effect of shRNA-induced knockdown of Gata2 and Tal1 for leukemia development in vivo. Upon transplantation, engrafted Gata2 and Tal1 knockdown LSK cells from Reg1Δ/Δ BM showed moderate LSK-HSC or CD34− HSC expansion compared to controls (Fig. 7c–f). Correspondingly, Gata2 and Tal1 down-regulation partially alleviated splenomegaly and mesenteric lymphadenopathy (Fig. 7g). We investigated whether Gata2 and Tal1 over-expression is involved in the observed cell cycle abnormalities of Regnase-1-deficient HSPCs. Cell cycle abnormalities observed in CD34− HSCs due to Regnase-1 deficiency were partially recovered by knock-down of Gata2 and Tal1 expression (Fig. 7h). In summary, our in vitro and in vivo data indicate that dysfunction of Regnase-1 causes hyperactive Gata2 and Tal1 function, resulting in exuberant self-renewal of HSPCs and aberrant differentiation.

## Discussion
HSPC homeostasis must be maintained by proper regulation of intracellular signaling for the continuous provision of blood cells throughout life. The cell fate of HSPCs, self-renewal, quiescence, and differentiation is decided by a network of transcription factors[34,35]. However, the regulatory mechanisms responsible for the intricate coordination of transcription factors in HSPCs according to the situation has not been investigated. Here, we identified Regnase-1 as a molecule expressed in HSPCs and inversely correlated with their proliferative capacity. A previous report had shown that Regnase-1 modulates immune responses by degrading inflammatory cytokine mRNAs in macrophages and T lymphocytes[14]. Using mice lacking Regnase-1 in hematopoietic cells, we report that this factor controls proliferation and differentiation of CD34− HSCs and is also involved in homeostasis of myelopoiesis. A significant expansion of Regnase-1-deficient CD34− HSCs was provoked through promoting cell cycle progression, accompanied by an aberrant expression of cyclin-dependent kinase inhibitory proteins. Importantly, Regnase-1-deficient mice exhibited abnormal hematopoiesis including a marked accumulation of abnormal blasts in the blood, lethal within three months of age. Mechanistically, we found that

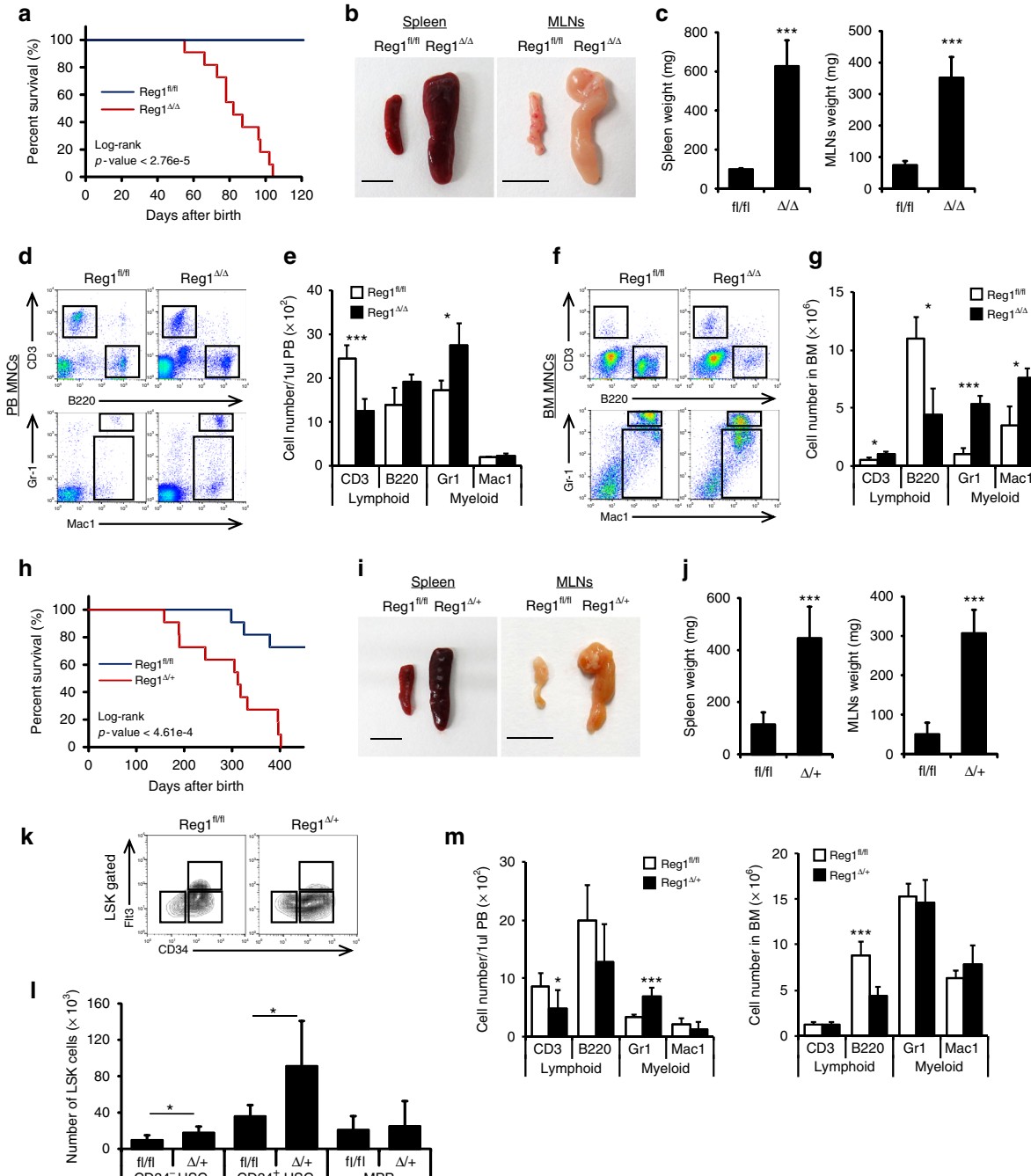

**Fig. 4** Loss of Regnase1 causes impaired hematopoietic functions. **a** Kaplan-Meier survival curve of control (fl/fl) and Regnase-1-KO (Δ/Δ) mice ($n = 11$). P values were calculated by the logrank test. **b** Representative gross appearance of the spleen and mesenteric lymph nodes (MLNs) collected from 8-week-old mice as indicated. The scale bars represent 1 cm. **c** Spleen (right) and MLNs (left) weight of 8-week-old mice as indicated ($n = 4$ per group). **d** Flow cytometric analysis of lymphocytes and monocytes in PBMCs from 8-week-old mice as indicated ($n = 3$ per group; 3 independent experiments). **e** Quantification of frequencies of lymphoid and myeloid cells in PBMCs. **f** Flow cytometric analysis of lymphocytes and monocytes in BM MNCs from 8-week-old mice as indicated ($n = 3$ per group; 3 independent experiments). **g** Quantification of frequencies of lymphoid and myeloid cells in bone marrow MNCs. **h** Kaplan-Meier survival curve of control (fl/fl) and Regnase-1 hetero-mutant (fl/Δ) mice ($n = 11$). P values were calculated by the logrank test. **i** Representative images of the spleen and MLNs collected from 12-month-old mice as indicated. The scale bars represent 1 cm. **j** Spleen (right) and MLNs (left) weight of 12-month-old mice as indicated ($n = 4$ per group). **k, l** Representative flow cytometric analysis and quantitative evaluation of LSK-HSC numbers in the BM of 8-week-old mice as indicated ($n = 3$ per group; 3 independent experiments). **m** Quantification of ratios of lymphocytes and monocytes in PB (left) and BM (right) MNCs from 12-month-old mice as indicated ($n = 3$ per group; 3 independent experiments). The ratio was calculated by flow cytometric analysis. Error bars indicate mean ± SD. *$p < 0.05$; **$p < 0.01$, ***$p < 0.005$, logrank test in **h**, two-sided $t$-test in **c, e, g, j, m, l**

Regnase-1 targets the 3′UTR of mRNAs encoding *Gata2* and *Tal1*, and post-transcriptionally modulates their activity through mRNA degradation. Excessive proliferation of HSPC resulting from Regnase-1 deficiency may allow progression to abnormal hematopoiesis by promoting mutagenesis. Collectively, our data reveal a role for Regnase-1 in regulating HSPC proliferation in hematopoietic homeostasis, the dysfunction of which could lead to the generation of leukemic cells.

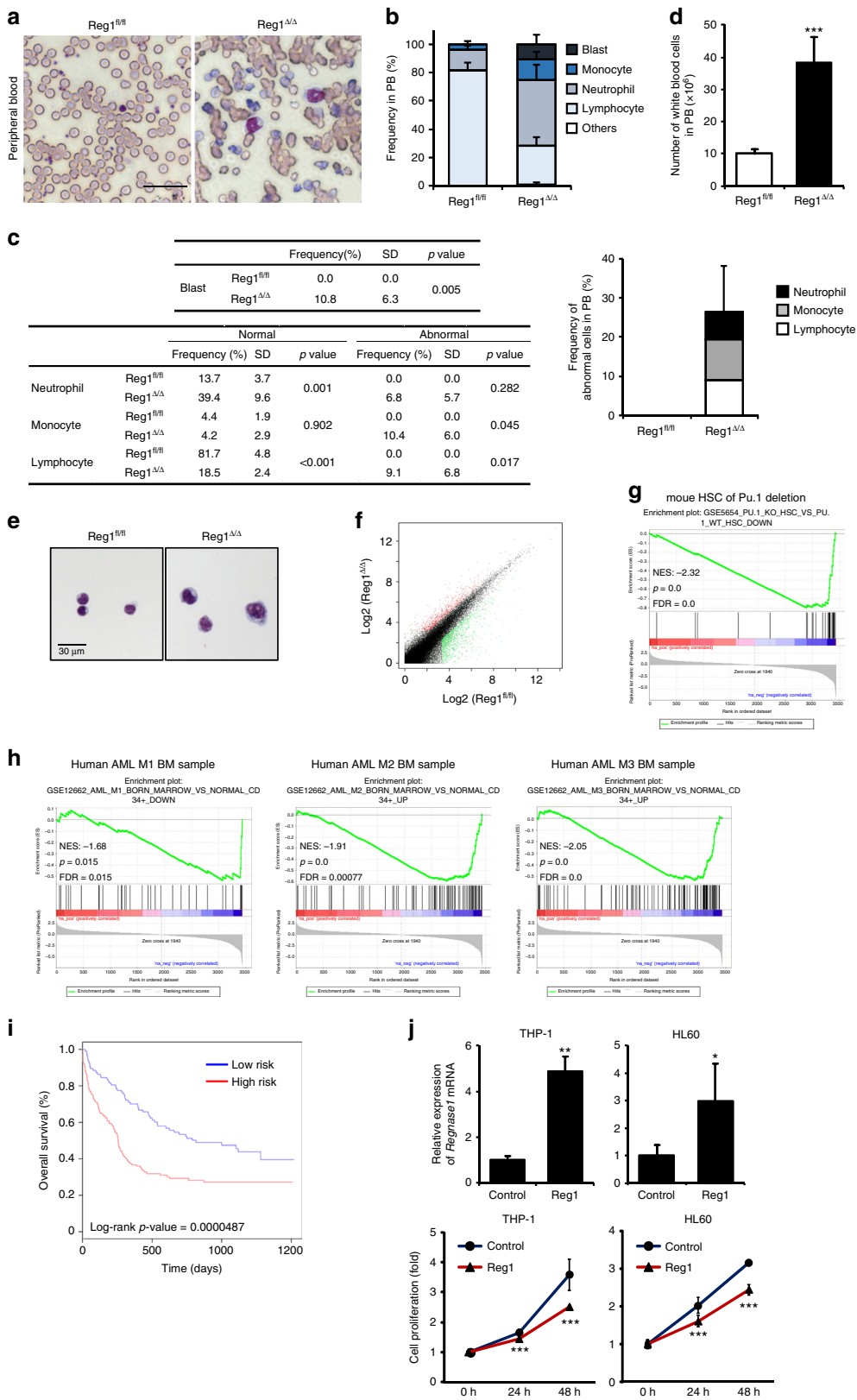

The fate of hematopoietic cells is controlled through transcription factor-mediated gene expression. Thus far, many transcription factors related to HSPC development have been identified, and their functional roles clarified by studies using genetically modified mice[36,37]. However, hematopoiesis in the living organism is not controlled by the simple presence or absence of transcription factors, but by intricately refined tuning of their activities. In addition, gene regulatory networks involving interactions of many different combinations of transcription factors with diverse activities ensure robustness and adaptability of HSPC function. In our study, we found that Regnase-1 degrades *Gata2* and *Tal1* mRNA, which encode transcription

**Fig. 5** Regnase-1-deleted HSPCs show a phenotype and gene expression profile similar to leukemia. **a** Representative field of Wright-Giemsa-stained BM smears of 8-week-old control (fl/fl) and Regnase-1-KO (Δ/Δ) mice. The scale bar shows 30 μm. **b** Quantification of frequencies of blasts and other MNCs in PB on Wright-Giemsa-staining (n = 3 per group; 3 independent experiments). **c** Quantification of abnormal cells in the PB. **d** Peripheral white blood cell counts in blood samples from 8-week-old control (fl/fl) or Regnase-1-KO (Δ/Δ) mice (n = 3 per group). **e** Wright-Giemsa staining of CD34⁻ HSCs. The scale bars represent 30 μm. **f** Scatter-plot representation of the transcriptional differences between control Reg1^flox/flox (fl/fl) and Vav1-iCre; Regnase-1^flox/flox(Δ/Δ) CD34⁻ HSCs. The color indicates >2-fold differential gene expression (n = 3 per group and n = 2 per group; 2 independent experiments; the average of the two is shown). **g** Gene set enrichment analysis comparing CD34⁻ HSCs from control (fl/fl) and Regnase-1-KO (Δ/Δ) mice with the association between genes upregulated following Regnase-1 deletion and Pu.1 deletion. **h** Human HSC signatures of AML compared with Regnase-1-deficient CD34⁻ HSCs. The normalized enrichment scores (NES), P-value and q-value (FDR) are indicated on each plot. **i** Kaplan-Meier plot of AML patient survival data based on similarity of gene expression profiles of loss of Regnase-1. P values were calculated by the logrank test. **j** *Regnase-1* mRNA expression (upper panel) and proliferation (lower panel) of THP1 and HL60 leukemic cells transfected with mouse Regnase-1 cDNA expression vector (n = 3 per group; 3 independent experiments). Error bars indicate mean ± SD. *p < 0.05; **p < 0.01, ***p < 0.005, log rank test in **i**, two-sided t-test in **c**, **d**, **j**

factors essential for HSPC integrity. Loss of function of Regnase-1 in HSPCs results in up-regulation of Gata2 and Tal1 expression and leads to excessive self-renewal. These findings suggest that Regnase-1 could have a role in fine tuning the transcription factor network by post-transcriptional regulation. Regnase-1 is a member of the CCCH zinc finger protein family harboring a PIN domain with RNAse activity. Several studies have demonstrated that Regnase-1 acts as a transcription factor regulating apoptosis-related gene expression and that it has a role in miRNA biogenesis by cleaving target pre-miRNAs[38,39]. Recently, it was reported that Regnase-1 has post-transcriptional regulatory activity through recognition of the specific 3′UTR sequence of the target mRNA and induction of mRNA degradation via its RNAse activity, but data on the binding sequence of Regnase-1 and mechanism of RNA cleavage are still limited[40,41]. In macrophages, Regnase-1 controls the immune response by degrading mRNAs encoding *IL-1β*, *IL-6*, and *IL-12p40* mRNA[16], whereas in T cells, it targets mRNAs encoding immunoregulatory proteins such as *c-Rel*, *OX40*, and *IL-2*[16]. Because the target mRNA varies depending on cell type, we showed that *Gata2* and *Tal1* degradation is specific for CD34⁻HSCs, although inflammatory cytokine expression was not significantly altered in Regnase-1-deleted HSPCs. Regnase-1 deficiency in immune cells such as macrophages and T-cells causes a systemic inflammatory and severe autoimmune disease phenotype[14,16]. Our results show that Regnase-1 deletion in HSPCs leads to their excessive proliferation and aberrant differentiation. Therefore, Regnase-1 regulates specific mRNA expression in each blood cell type and functions in a context-dependent manner. On the other hand, because Cre-mediated Regnase-1 deletion occurred not only in HSPCs but also lineage-committed cells in Vav1-iCre; Regnase-1^flox/flox mice, phenotypes in mutant mice are not exclusively caused by abnormal HSPCs. The cell type-specific regulatory mechanism of Regnase-1 might involve cell-specific functions of this factor. During an inflammatory response, Regnase-1 expression is regulated by Toll-like receptor-mediated and pro-inflammatory signals in macrophages or T-cell receptor-mediated signals in T-cells[16]. For HSPC maintenance, the regulatory mechanism of expression and function of Regnase-1 has remained unclear. Further precise analysis may unmask regulatory mechanisms responsible for Regnase-1 activity by analyzing the signals from the BM microenvironmental niche.

The transcription factors Gata2 and Tal1/SCL are expressed in HSPCs and are implicated in hematopoietic differentiation and proliferation[42,43]. Either reduction or gain of Gata2 function caused by genetic alteration leads to abnormal hematopoiesis and predisposes to AML. Lower amounts of Gata2 caused by mutations contribute to the development of myelodysplastic syndrome (MDS) and AML[44]. Gata2 is highly expressed in AML and its overexpression through p38/ERK-dependent signaling portends a poor prognosis of cytogenetically normal AML[45,46]. Gene

knockdown or pharmacological inhibition of Gata2 in AML cells can relieve drug resistance by inducing an increase of the apoptotic population. (Yang, L., Sun, H., Cao, Y. *PLoS One* (2017)). Tal1 is also known to act as a proto-oncogene, and aberrant Tal1 transcription is related to the pathogenesis of T cell acute lymphoblastic leukemia (T-ALL)[47,48]. Aberrant expression of Tal1 in T-ALL represses the expression of genes controlling cell homeostasis[49,50]. We found that deletion of Regnase-1 in HSPCs causes aberrant differentiation by inducing overexpression of molecules that maintain normal hematopoiesis, such as Gata2 and Tal1. In our experiments, however, knockdown of Gata2 and Tal1 in Regnase-1-deleted HSPCs was not sufficient to fully normalize hematopoiesis. Therefore, other Regnase-1 target mRNAs are likely to exist. In addition, no direct binding of Gata2 or Tal1 to p21 or p57 promoter regions was found using chromatin immunoprecipitation assays (Supplementary Fig. 7b); thus, further exploration of the regulatory mechanisms of p21 and p57 expression by Gata2 and Tal1 is required. Further work is required to unveil the mechanism of leukemogenesis.

Regnase-1 acts as a fine-tuning regulator of global transcription factors not only for physiological hematopoiesis but also for leukemogenesis. Emerging evidence from this research using Regnase-1 knock-out mice suggests that this molecule has potential relevance for human AML. Bioinformatics analysis revealed that gene expression sets of Regnase-1-deleted mouse HSPCs were significantly enriched in the group of human AMLs tested, but gene mutations and mRNA alterations of *Regnase-1* in human AML patients had not been reported thus far. Therefore, Regnase-1 expression may be regulated by protein modification such as phosphorylation and ubiquitination. Regnase-1 is a powerful tumor suppressor gene, decreased amounts of the product of which results in the spontaneous development of abnormal hematopoiesis even in heterozygous mice. This may provide an opportunity for drug discovery for the treatment of AML. Previous studies in CD4⁺ T cells showed that Regnase-1 expression and function is suppressed by MALT1, which possesses an arginine-specific protease activity[16,51]. MALT1 inhibitors are therefore possible candidates for enhancing Regnase-1 expression, suggesting their potential as therapeutic agents in inflammatory diseases and viral infections[52,53]. For therapeutic applications, it will be necessary to elucidate the molecular mechanisms which could be employed to enhance or repress Regnase-1 activity in leukemic cells.

In summary, here we have shown that Regnase-1 contributes to hematopoietic homeostasis by post-transcriptional regulation of target mRNA stability. Regnase-1 may simultaneously modulate the expression of numerous essential factors for HSPC self-renewal and differentiation including Gata2 and Tal1. We suggest that Regnase-1, a post-transcriptional regulator, is a causative molecule for the development of abnormal hematopoiesis, and that understanding of the regulatory mechanisms controlling

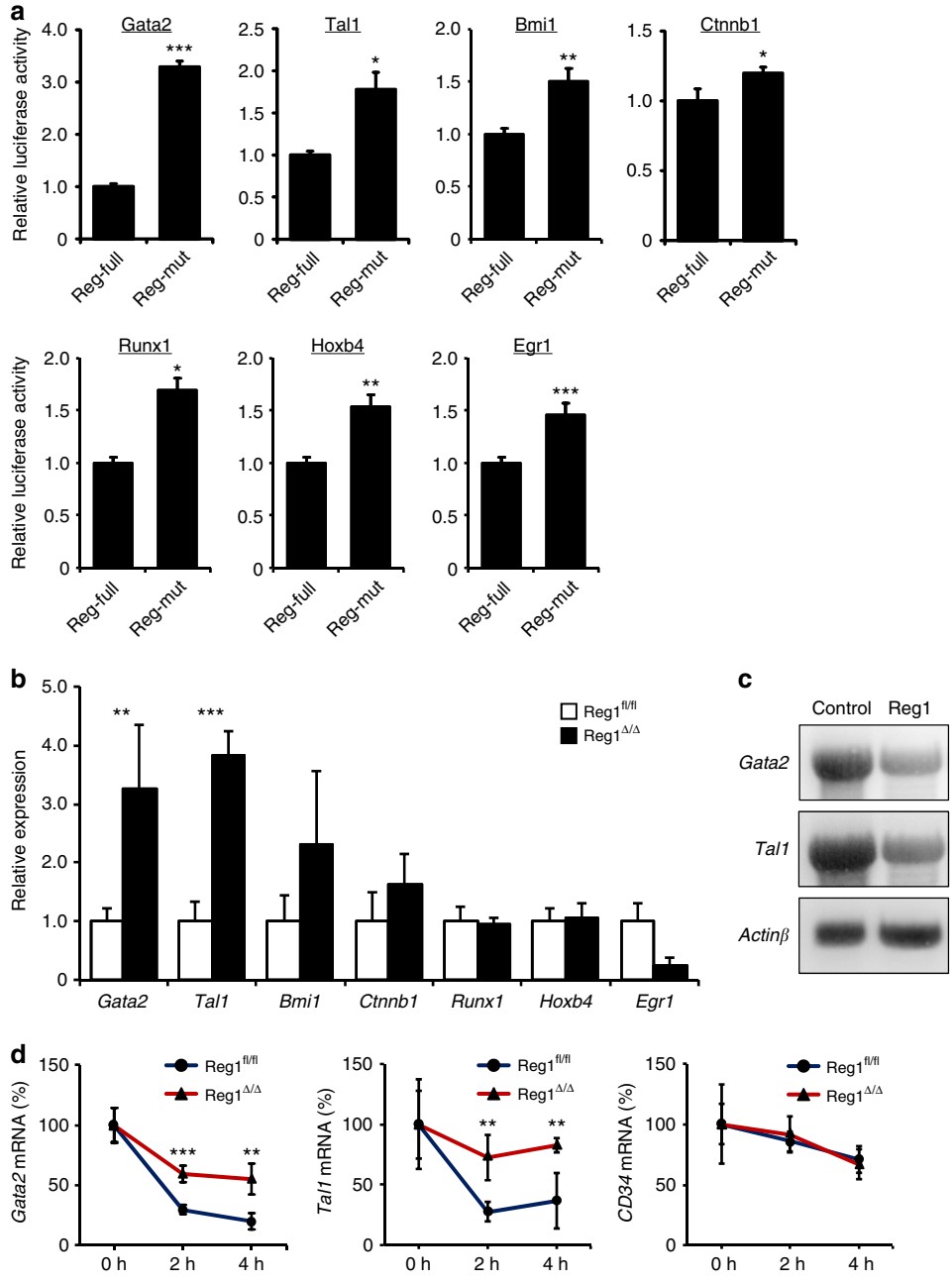

**Fig. 6** Regnase-1 recognizes and degrades *Gata2* and *Tal1* mRNA. **a** Luciferase activity of HEK293 cells transfected with luciferase reporter plasmids containing 3′UTRs of the indicated genes and either *Regnase-1* expression plasmid or inactive *Regnase-1* mutant plasmid (3 independent experiments). **b** qRT-PCR analysis of candidate gene expression in CD34⁻ HSCs from control (fl/fl) or Regnase-1-KO (Δ/Δ) mice BM ($n = 3$ per group; 3 independent experiments). **c** THP1 cells were transfected with human *Regnase-1* cDNA expression vector or control (empty) vector. After 48 h, levels of *Gata2*, *Tal1*, and *Actin β* mRNA were determined by Northern blotting. Representative data of two independent experiments are shown. **d** Quantitative RT-PCR of *Gata2*, *Tal1*, and *CD34* expression in CD34⁻ HSCs from control (fl/fl) or Regnase-1-KO (Δ/Δ) mouse BM treated for 0–4 h with actinomycin D (ActD) ($n = 3$ mice per group). Data are expressed as fold-change relative to 0 h treatment. Error bars indicate mean ± SD. *$p < 0.05$; **$p < 0.01$, ***$p < 0.005$, two-sided *t*-test in **a**, **b**, **d**

Regnase-1 expression and identifying additional target genes will provide valuable information for therapies targeting AML.

## Methods

**Mice**. C57BL/6 mice (B6-Ly5.2) were purchased from Japan SLC (Shizuoka, Japan) and C57BL/6-Ly5.1 mice were purchased from Sankyo Labo Service (Tsukuba, Japan) and used between 6 and 12 weeks of age. Regnase-1 floxed mice were generated as previously described[16]. A2Kio/J (Vav1-iCre) mice were purchased from The Jackson Laboratory (Bar Harbor, ME). Animals were housed in

environmentally controlled rooms of the animal experimentation facility at Osaka University. All experiments were carried out under the guidelines of Osaka University Committee for animal and recombinant DNA experiments and were approved by the Osaka University Institutional Review Board.

**Cell culture and transfections**. Human leukemia cell lines (HL60 and THP-1) and HEK293 cells were provided by the RIKEN cell bank. Cells were maintained in RPMI-1640 medium (SIGMA) or DMEM medium (SIGMA) supplemented with 10% heat-inactivated fetal bovine serum (SIGMA). Transfections of Regnase-1

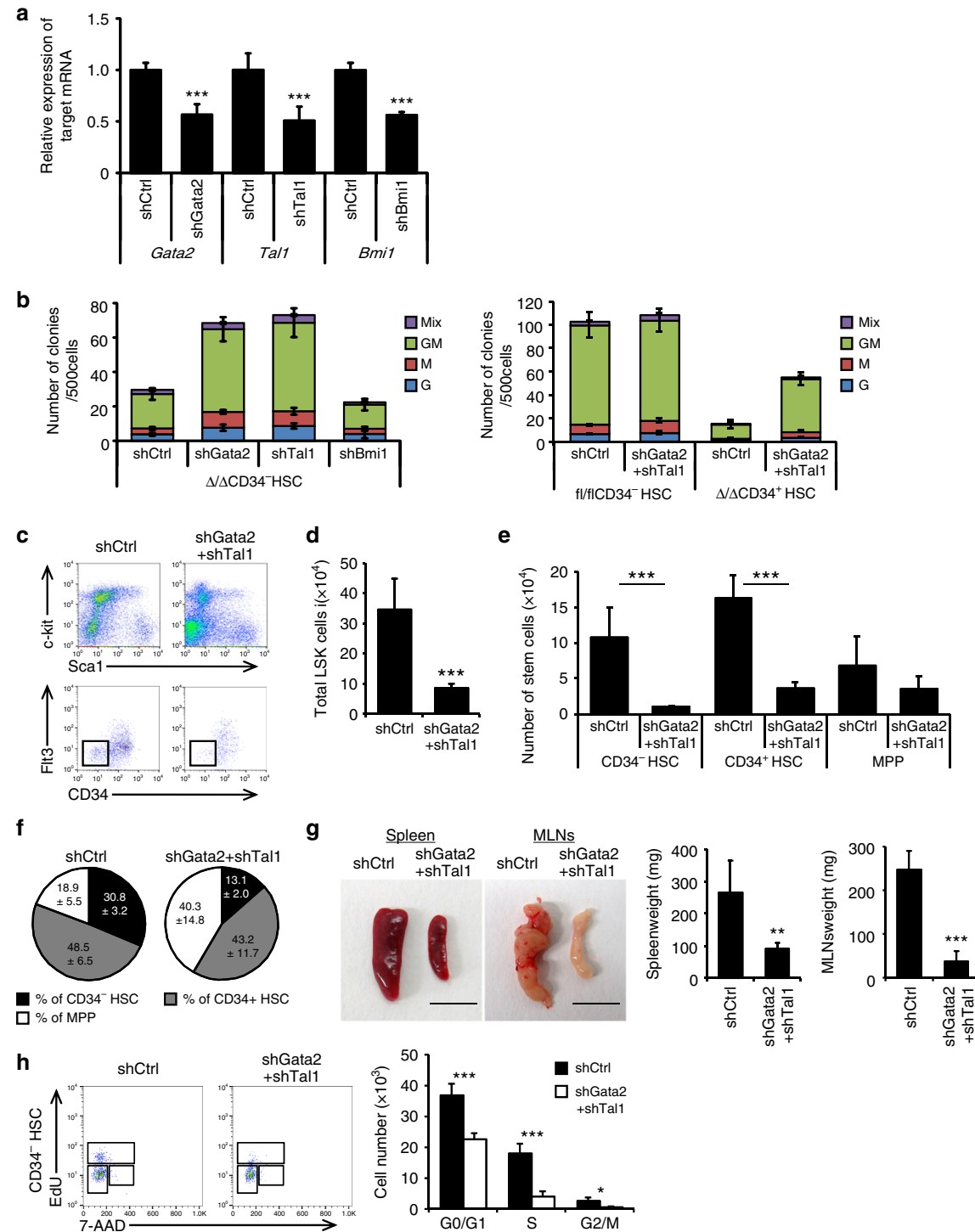

expression vector[33] or empty vector (pcDNA3) were performed using FuGENE HD transfection reagent (Promega) according to manufacturer's instructions. Proliferation of HL60 and THP-1 cells was evaluated using Cell Counting Kit-8 (Dojindo). All of the cells were authenticated by examination of morphology and growth characteristics, and were confirmed to be mycoplasma-free.

**Immunohistochemical analysis**. Tissue fixation and staining of sections or tissues with antibodies was performed by the following method. Freshly-excised femoral or tibial bone from WT mice was embedded in super cryoembedding medium (Leica Microsystems) and frozen in liquid nitrogen. Next, 10-μm cryosections were generated via the Kawamoto film method[54]. An anti-c-Kit (AF1356, R&D Systems, dilution 1/100), anti-Sca1 (553333, BD Biosciences, dilution 1/100), anti-CD71 (113802, Biolegend, dilution 1/100) and, anti-Regnase-1 antibodies (dilution 1/100)[16] were used for staining[55]. Cell nuclei were visualized with TO-PRO-3 and sections were

examined by confocal microscopy (TCS/SP8; Leica). EdU labeling and staining was performed using a Click-iT® Plus EdU Alexa Fluor 555 Imaging Kit (Thermo Fisher) according to the manufacturer's instructions. In all assays, isotype-matched Ig was used as a negative control and it was confirmed that the positive signals were not derived from nonspecific background. Images were processed using Photoshop CS2 software (Adobe Systems). All images shown are representative of 3 to 5 independent experiments.

**Flow cytometric analysis and cell sorting**. Flow cytometry and isolation of blood cells from fetal liver and BM was performed by the following method. Briefly, fetal livers dissected from E14.5 embryos or BM cells obtained from femur and tibia were dispersed into single-cell suspension by gentle pipetting[55]. Fluorescence-labeled anti-CD34 (12-0349-42, eBioscience, dilution 1/100), -Flk2/CD135 (135309, dilution 1/100), -Sca-1 (108127, dilution 1/100), -c-Kit (105825, dilution

**Fig. 7** The effect of Regnase-1 on HSPCs is mediated through Gata2 and Tal1. **a** Relative expression of *Gata2*, *Tal1*, and *Bmi1* in CD34⁻ HSCs infected with shRNA virus targeting *Gata2*, *Tal1*, *Bmi1* or scrambled control (shCtrl). Bmi1 was used as a non-candidate negative control. Data are expressed as fold-change relative to control shRNA infection (*n* = 3 per group). **b** Colony-forming assay of Regnase-1-deficient CD34⁻ HSCs infected with either an shCtrl, shGata2, or shTal1 shRNA virus (left), and control (fl/fl) or Regnase-1-KO (Δ/Δ) CD34⁻ HSCs infected with control or shGata2+ shTal1 shRNA virus (right). G indicates CFU-granulocyte; GM CFU-granulocyte/monocyte; M CFU-monocyte; E CFU-erythroid; and mix mixed CFU-granulocytes, monocyte, erythroid, and megakaryocyte (*n* = 4 per group; 4 independent experiments). **c** Regnase-1-KO (Δ/Δ) CD34⁻ HSCs infected with shCtrl or shGata2+ shTal1 shRNA virus were transplanted and analyzed by flow cytometry (*n* = 3 per group; 3 independent experiments). **d–f** Numbers and populations of LSK-HSCs or CD34⁻ HSCs in total BM assessed by flow cytometry in **c**. Data represent the means ± SD. **g** Representative gross appearance of the spleen and mesenteric lymph nodes (MLNs) collected from bone marrow-transplanted mice as shown in **c**. The scale bars represent 1 cm. **h** Cell-cycle analysis of control BM CD34⁻ HSCs in **c** by flow cytometry using EdU/7AAD staining. Dot plots indicate the frequency of CD34⁻ HSCs in G0/G1 (EdU⁻ 7AAD⁻), G2/M (EdU⁺), or S (EdU⁺ 7AAD⁺) phase of the cell cycle (*n* = 3 per group; 2 independent experiments). Error bars indicate mean ± SD. *$p < 0.05$; **$p < 0.01$, ***$p < 0.005$, two-sided *t*-test in **a**, **d**, **e**, **g**, **h**. Regnase-1 is known to mediate post-trasncriptional regulatory activity through degradation of target mRNAs. Here, the authors show that Regnase-1 regulates self-renewal of haematopoietic stem and progenitor cells through modulation of the stability of Gata2 and Tal1 mRNA

1/100), -FcγR (101325, dilution 1/100), -IL7Rα (121111, dilution 1/100), -CD71 (113807, dilution 1/100), -CD3 (100205, dilution 1/100), -Mac1 (101205, dilution 1/100), -B220 (103205, dilution 1/100), -Gr1 (108407, dilution 1/100), -CD45 (103134, dilution 1/100), -CD93 (136503, dilution 1/100), -CD150 (115909, dilution 1/100), -CD48 (103403, dilution 1/100), -CD45.1 (110730, dilution 1/100), -lineage Cocktail antibodies (133301, all purchased from BioLegend, dilution 1/30) were used. Stained cells were sorted by FACS Aria (BD Biosciences) and analyzed with FlowJo software (TreeStar). Gating strategies are shown in each figure and Supplementary Fig. 8. For RNA expression analysis and RNA-seq experiments, mRNA was directly isolated from sorted cells. For Giemsa staining, sorted cells were spun down onto glass slides using a Cytospin 4 instrument (Thermo Scientific) and stained with May-Grünwald-Giemsa (Muto Pure Chemicals).

**Quantitative real-time PCR (qRT-PCR)**. Target gene mRNA levels were quantified by qRT-PCR or Western blotting as described[56]. PCR primers used in this work are described in Supplementary Data 2.

**Colony-forming cell (CFC) assay**. CFC assays were performed by culturing LSK cells ($1 \times 10^3$/dish) or CD34⁻ HSCs (CD34⁻ Flk2-LSK) ($1 \times 10^3$/dish) collected from BM in triplicate 35-mm petri dishes (FALCON) containing 1 ml MethoCult GF M3434 medium (Stem Cell Technologies). After 10 days of incubation at 37 °C in 5% CO₂ in air, colony forming unit–granulocyte monocyte (CFU-GM), colony forming unit-granulocyte (CFU-G), colony forming unit-monocyte (CFU-M), and colony-forming unit mix (CFU-Mix) were counted.

**Cell-cycle analysis and apoptosis assay**. The proportion of HSPCs in G0/G1, G2/M, and S phases was analyzed using EdU incorporation assay over 24 h using Click-it EdU Alexa Fluor 647 Cell Flow Cytometry Assay Kit and Click-iT Plus EdU Alexa Fluor 488 Flow Cytometry Assay Kit (Thermo Fisher Scientific) according to the manufacturer's instructions. DNA content was detected with 7-AAD. To distinguish between cells in G0 and G1 phases, BM cells were stained with Hoechst 33342 at 37 °C. After 45 min, 1 µg/ml PY was added and cells were incubated at 37 °C for 45 min[57]. Samples were analyzed by FACSAria flow cytometry. Apoptotic cells were analyzed using the APC-Annexin V/Dead Cell Apoptosis Kit (Invitrogen) according to the manufacturer's protocol. Briefly, BM cells were treated with reagents provided in the kit and analyzed by FACS Aria flow cytometry and quantified using the FlowJo software.

**Competitive transplantation assay**. CD34⁻ HSCs (500 cells) from Reg1^flox/flox or Reg1^Δ/Δ mice were mixed with wild-type CD45.1⁺ mice BM cells ($5 \times 10^5$) and transplanted intravenously into lethally irradiated (8 Gy) CD45.1⁺ recipient mice. Sixteen weeks after transplantation, we assessed chimerism of donor-derived HSPCs and peripheral blood (PB) in recipient mice.

**Myelosuppression models**. Mice were intraperitoneally administered 5-fluorouracil (5-FU; Kyowa Hakko) at a dose of 250 mg/kg body weight once per week for 2 weeks and their survival was monitored daily in the latter week. Bone marrow cells were collected and analyzed after 0, 2, 4, 5, and 6 days of administration.

**Hemoglobin measurement**. Hemoglobin concentration was quantified using a Hemoglobin Assay Kit (Biochain) according to the manufacturer's instructions.

**Northern blotting**. Total cellular RNA was prepared from THP1 cells transfected with a Regnase-1 expression plasmid or a control plasmid using RNeasy mini kits (Qiagen). RNAs were resolved by electrophoresis on a formaldehyde/agarose gel and transferred onto a positively charged nylon membrane (GE Healthcare). Probe

preparation and hybridization were conducted with the DIG Northern Starter Kit (Roche Diagnostics), according to the manufacturer's protocol. DIG-labeled RNA probes targeting the *Gata2* and *Tal1* gene were constructed by T7 RNA polymerase reaction using the primer sets described in Supplementary Data 2. The human *β-actin* RNA probe was used to normalize the amount of RNA bound to the membrane.

**Stability of mRNA**. CD34⁻ HSCs sorted from control (fl/fl) or Regnase-1-KO (Δ/Δ) mouse BM were cultured for 1 h. Actinomycin D (SIGMA) at 5 µg/ml was added to the culture medium to terminate transcription, and total RNA was harvested at the indicated time[16]. RNA was subjected to qRT-PCR to determine *Gata2*, *Tal1*, and *CD34* levels.

**Chromatin immunoprecipitation (ChIP) assay**. The ChIP assay was performed using the SimpleChIP Plus Enzymatic Chromatin IP Kit (Cell Signaling Technology), according to the manufacturer's protocol. Chromatin solutions were prepared from THP1 cells transfected with a Regnase1 expression plasmid or a control plasmid, digested with micrococcal nuclease and sonicated to give the desired fragment length. Immunoprecipitation was performed overnight at 4 °C using an anti-Gata2 antibody (ab22849, Abcam, dilution 1/200) or anti-Tal1 antibody (C15200012, Diagenode, dilution 1/200). PCR analysis was then performed using the primer sets described in Supplementary Data 2. The specificity of the assay was validated by normal IgG as the negative control.

**Luciferase assay**. Validation of Regnase-1 target genes by luciferase assay was performed as described[33]. Fragments of the 3′UTR of the mRNA of interest were amplified with PCR using a specific primer (Supplementary Data 2) and inserted into the pGL3-luciferase plasmid. HEK293 cells were co-transfected with pGL3-3′ UTR plasmid or pGL3-empty plasmid together with Regnase-1 expression plasmid or empty control plasmid. After 48 h, cells were lysed and luciferase activity determined using the Dual-Luciferase Reporter Assay System (Promega). Renilla luciferase was transfected simultaneously and served as an internal control.

**Retroviral vector construction and transduction**. For knockdown experiments, oligonucleotides designed to target Gata2, Tal1, or Bmi1 (described in Supplementary Data 2) and scrambled oligonucleotides supplied by the manufacturer were annealed and cloned into the RNAi-ready pSIREN-RetroQ-zsGreen vector (Takara). Retroviruses were produced using PlatE packaging cells transfected with different plasmids as previously described[58]. Retrovirus supernatants were harvested 48 post-transfection and concentrated using the Retro-X Concentrator (Clontech). For transduction, $5 \times 10^4$ CD34⁻ HSCs sorted by flow cytometry were incubated for 72 h in cytokine conditioned medium (1% FCS, 100 ng/mL TPO, 100 ng/mL SCF, and 0.4 ng/ml IL-3) supplemented with virus particles. After transduction, cells were collected and used for CFC assays or bone marrow transplantation.

**cDNA libraries and RNA sequencing**. RNA samples isolated from CD34⁻ HSCs from control Reg1^flox/flox and Reg1^Δ/Δ (Vav1-iCre; Regnase−1^flox/flox) mice were reverse transcribed and amplified using the SMARTer Ultra Low Input RNA Kit for Sequencing–v3 (Takara Bio. Cat. 634849). Illumina sequencing libraries were constructed using Nextera XT DNA Librart Prep Kit (Illumina). After evaluating the quality and quantity of the constructed RNA-Seq libraries using a BioAnalyzer (Agilent Technologies), sequencing was performed on the HiSeq2500 platform with a 101-base paired-end read. Generated RNA-Seq tags were mapped to the reference mouse genome (GRCm38.p4) using STAR v2.4.2a[59]. We used the GENCODE annotation file (release version M17) for the mouse which can be downloaded from http://www.gencodegenes.org/. The tag counts of the first and last exons including 5′-untranslated and 3′-untranslated regions for each transcript

were calculated, and then normalized by the lengths of the exons. The transcripts per million (TPM) values for each transcript were also calculated with Salmon v0.9.1[60].

**Identification of key factors by bioinformatics analysis.** The microarray data set GSE69760 from BM and fetal liver Lineage[−] Sca-1[+] c-Kit[+] (LSK) HSPCs in developmental stages were downloaded from Gene Expression Omnibus. To identify the crucial factors in regulation of self-renewal and differentiation of HSPCs, we screened genes according to the following criteria: (1) Expression level in adult BM was >2. (2) Expression level in adult BM was >10 times that of HSPCs from E14.5 FL. (3) Gene encodes intercellular proteins. (4) Gene encodes a cytokine, enzyme, G-protein coupled receptor, growth factor, ion channel, kinase, ligand-dependent nuclear receptor, peptidase, phosphatase, transcription regulator, translation regulator, transmembrane receptor, or transporter which are categorized in the Ingenuity Knowledge Base (https://www.qiagenbioinformatics.com/products/ingenuity-pathway-analysis). After filtering, 10 genes were selected in Fig. 1a.

**Gene set analysis.** By analyzing RNA-seq gene expression profiles of Vav1-iCre; Reg1[flox/flox] mice (Reg1[Δ/Δ], case) and 8-week-old control Reg1[flox/flox] mice (Reg1[flox/flox], control), differential expression analysis was performed with the Bioconductor R package limma[61]. A total of 801 up-regulated genes in the Reg1[Δ/Δ] animals was detected according to the following criteria: log2-fold change >2× the average expression value in Reg1[Δ/Δ] (case) compared with Reg1[flox/flox] (control); average expression value in the case was >90th percentile of the case, and the FDR value was <10[−3]. Reciprocally, 1179 down-regulated genes in Reg1[Δ/Δ] animals were detected according to the following criteria: log2-fold change <1/2 of the average expression value in cases compared with controls; average expression value in the control was >90th percentile of the control, and the FDR value was <10[−3]. Supplementary Data 3 in this differential expression analysis. To identify gene sets enriched among these down-regulated genes we used the Metascape analyses from the web site http://metascape.org/[62]. The results are summarized in Supplementary Data 4. We also performed gene set enrichment analysis (GSEA) for a ranked list of genes using log2-fold change[63]. Based on the GSE12662 data, we identified genes with high or low expression in AML (18 M1, 19 M2, and 14 M3 patients) relative to normal bone marrow cells ($n = 5$) by the following method: (a) up-regulated genes with a case mean/control mean ratio >2, $t$-test $p$-value <1e−3, and case mean expression ≥75th percentile of mean expression; (b) down-regulated genes with a case mean/control mean ratio <1/2, $t$-test $p$-value <1e−3, and control mean expression ≥75th percentile of mean expression. Gene expression data of Regnase-1-deficient mice were correlated with orthologous human genes, and whether the AML signature genes were coordinately expressed in LT-HSCs from Regnase-1 deficient mice was assessed by GSEA. For our analyses, we used our predefined gene sets which are related to leukemia (Supplementary Data 5) and we considered gene sets with FDR < 0.1 and $p$-value < 0.05 as significant. The results of the GSEA are summarized in Supplementary Data 6.

**Survival analysis based on Regnase-1 target gene expression.** To construct the prognostic model for AML patients based on the expression profiles of 801 target genes directly or indirectly down-regulated by Regnase-1 categorized as hematopoietic genes in the Ingenuity Knowledge Base, we downloaded the microarray expression data set GSE12417 from the Gene Expression Omnibus (GEO) database. We then used a Lasso-regularized Cox proportional hazard model with the glmnet package (version 2.0–5) (ref. [64]) in the R statistical environment (version 3.3.2) to build a Regnase-1 signature-based prognostic classifier. Among the 801 genes in mouse, 242 genes were mapped to that orthologous genes of human in this dataset by using the Ingenuity Knowledge Base. The tuning parameter in the Lasso regularization was chosen by cross-validation. Using the prognostic model, patients were classified into two groups based on whether the risk score in the Cox model was more than 0 (high risk and worse prognosis) or less than 0 (low risk and better prognosis). To evaluate the prognostic significance of the Cox model, we used the Kaplan-Meier method and the $p$-value was calculated using the log-rank test.

**Statistical analysis.** All data are presented as the means ± s.d. None of the data were excluded from the analyses. Investigators were not blinded to allocation during cell and mouse assays. For in vivo studies, mice of the same age and similar weight were randomly assigned to experimental or control groups. For statistical analysis, the statcel4 software package (OMS, Saitama, Japan) was used, with analysis of variance performed on all data, followed by the Tukey–Kramer multiple comparison test. When only two groups were compared, a two-sided Student's $t$-test was used.

**Reporting summary.** Further information on experimental design is available in the Nature Research Reporting Summary linked to this article.

## Data availability
The authors declare that all data supporting the findings of this study are available within the article and its Supplementary information. Any other data are available from the authors upon reasonable request. The RNA-seq data used in this manuscript are available in the GEO under the accession code GSE125546.

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

## Acknowledgements

We thank Dr. O. Takeuchi (IFLMS, Kyoto University, Japan) for supplying mutant mice, Dr. D. Okuzaki (RIMD, Osaka University, Japan) for performing RNA-seq analysis and Mr. R. Sasaki (Microeyes, Japan) for his help with the microscopic analysis. We are also grateful to Ms. S. Urakami, Ms. M. Omiya, Ms. Y. Mori, and Ms. N. Fujimoto for technical assistance. This work was supported by Grant-in Aid for Young Scientist A (No. 16H06147), Grant-in-Aid for Challenging Exploratory Research (15K14380) from the Japan Society for the Promotion of Science (JSPS), Takeda Science Foundation, SGH Foundation, the Kanae Foundation for the Promotion of Medical Science, the Uehara Memorial Foundation, Inamori Foundation, Ichiro Kanehara Foundation, the Mochida Memorial Foundation for Medical and Pharmaceutical Research, the Princess Takamatsu Cancer Research Fund, Osaka Cancer Society Research Grant, the Daiichi-Sankyo Foundation of Life Science, the Nakajima Foundation, the Japan Foundation for Applied Enzymology, Japan Agency for Medical Research and Development (AMED) under Grant number (18cm0106508h0002, 18gm5010002s110) and Japan Society for the Promotion of Science (JSPS) Grants-in-Aids for Scientific Research (A) (16H02470).

## Author contributions

H.K. and N.T. designed and performed most of the experiments. F.M., W.J., T.Sa., Y.H., H.N., Y.K., F.A., M.S., Y.S., T.O., and S.A. supervised assays and performed some experiments. T.Sh. analyzed and interpreted data. H.K and N.T. analyzed results and wrote the manuscript.

## Additional information

**Competing interests:** The authors declare no competing interests.

