## [Peer Review File · Nature Communications]

Reviewers' comments:

Reviewer #1 (Remarks to the Author):

In this study, the authors found that Regnase-1 is highly expressed in adult hematopoietic stem cells. Regnase-1 knockout in hematopoietic cells results in expansion of hematopoietic progenitors and reduction of quiescent stem cells specifically in bone marrow. Transplantation analyses show that Regnase-1 deficiency impairs repopulation potential of hematopoietic stem cells. These results indicate that Regnase-1 is required for stem cell functions. Cell cycle is activated in Regnase-1-deficient stem cells. Regnase-1 homozygous and heterozygous knockout mice exhibit splenomegaly and lymphadenopathy. Lymphoid cells are decreased and myeloid cells are increased in the Regnase-1 homozygous and heterozygous knockout mice, suggesting that Regnase-1 is indispensable for normal hematopoiesis. Blasts and cells with abnormal morphology were observed in the peripheral blood and bone marrows of Regnase-1-deficient mice. Gene expression profiles of Regnase-1 are similar to those of human hematopoietic stem cells in leukemia patients but not healthy people. Based on these results, the authors insist that Regnase-1 deficiency causes leukemia development. Finally, the authors performed analysis of RNA-seq database and identified Gata2 and Tal1 mRNAs as direct targets of Regnase-1, which account for hematopoietic phenotypes of Regnase-1-deficient mice.

This study provides an interesting insight according to Regnase-1 function in hematopoiesis, especially hematopoietic stem cells. Most of the experiments are carefully done. However, this reviewer feels that the present data is not sufficient to support the authors' conclusion. To support the authors' conclusions, there are several issues that should be addressed:

1. In Figure 1, the authors analyzed Regnase-1 expression levels of stem cells and progenitors. The authors should examine Regnase-1 expression levels in lineage-committed cells (e.g. B cells, T cells, myeloid, erythroid and megakaryocyte lineages), because the authors showed lineage-skewing in Regnase-1-deficient mice in Figure 4.
2. In Figure 4, the authors show increased myelopoiesis and decrease in lymphopoiesis in Regnase-1-deficient mice. Are they consequences of Regnase-1 deficiency in HSC (not lineage-committed cells)? As the authors mentioned in Discussion, Regnase-1 has cell type-specific function. The authors should discuss this issue.
3. Furthermore, this reviewer wonders whether Regnase-1 knockout affects erythropoiesis and megakaryopoiesis. Splenomegaly were often observed when abnormal erythropoiesis occurred. The authors should examine erythroid cells and megakaryocytes in the Regnase-1-deficient mice.
4. In Figure 5, the authors insist that Regnase-1-deficient mice develop leukemia. How was absolute number (concentration) of blasts in peripheral blood? Is this leukemia transplantable?
5. The authors identified Gata2 and Tal1 as responsible genes. Are the cell cycle-related genes p21 and p57 shown in Figure 3 direct targets of GATA2 or TAL1?

Reviewer #2 (Remarks to the Author):

Kidoya et al # NCOMMS-17-23012-T - Regnase-1-mediated post-transcriptional regulation is essential for hematopoietic stem cell homeostasis

These authors address a potential role for Regnase-1, a CCCH zinc finger RNA binding protein and

RNase, in the regulation of hematopoiesis. Their major finding is that “Regnase-1 regulates self-renewal of HSC through modulating the stability of Gata2 and Tal1 mRNA.” They also found that “dysfunction of Regnase-1 leads to rapid onset of AML”, and concluded that “Regnase-1-mediated post-transcriptional regulation is required for HSC maintenance and suggest that it represents a novel type of leukemia tumor suppressor.”

I will leave it to other reviewers to focus on the validity of the findings relating to the behavior of hematopoietic stem cells in the mutant mice, as well as the importance of these findings to the overall field of hematopoiesis regulation. My comments will focus on the RNA-Seq data and the conclusions regarding direct effects on Gata2 and Tal1 mRNA stability.

Despite going back and forth between the text, the methods section, and the figure legends, I was not able to figure out several aspects of the RNA-Seq data. It is possible that these pieces of information were hidden in other places, but I was not able to find them. I will list here some of my major problems with the information presented, but an overall comment is that it would be very helpful for the reviewer to have all the relevant information in one place, for example, in the methods or results section. Specific issues include, but are not limited to:

1. How were the HSC treated or cultured after the FACS separation? To my knowledge, we have no information on that.
2. How many biological replicates were used to generate the data in Fig. 5f and Fig. 6, showing the overall differential expression, i.e., how many mice were used to prepare how many independent cultures of HSC used in this experiment?
3. Regardless of the answers to 2 above, was the RNA-Seq experiment performed more than once?
4. What was the statistical method used to determine statistical significance of the differences?
5. Were the transcript data limited by an abundance cutoff?
6. Although the researchers took their gene sets through many screens before settling on the handful that they analyzed further as potential direct targets of Regnase-1, I think the reader should be given easy access to the actual gene lists, showing fold changes, an indication of abundance, and an indication of the significance of the fold changes in the original normalized RNA-Seq data. Perhaps a supplemental Excel file could be used to address most or all of these questions, as well as certain fundamental questions like the relative abundance of housekeeping transcripts, the abundance of the knocked out gene (reflecting cell purity), etc. In my view, all of this information is necessary to help the reader determine the validity of the authors' conclusions on this topic.
7. The authors then moved to their analysis of potential target transcripts. Rather than starting from the total collection of up-regulated transcripts, which most likely included many transcripts regulated by Regnase-1 either directly or secondarily, the authors moved to their exon gradient analysis, and limited the data still further by screening for up-regulated transcripts that are known to be involved in hematopoiesis. They thereby eliminated from the analysis many potential target transcripts that might well be involved in cellular function because they did not meet those screening criteria.
8. For Fig. 6c, I assume that these data were from real-time QPCR, but it is not clear that this is the case from the Results section.
9. It would be very helpful in the section on target transcript analysis to review the binding and/or cleavage sites for this protein, and the evidence to support these.
10. I don't think the co-transfection assays with luciferase readouts are adequate demonstrations that the mRNAs studied are direct targets in vivo. It would not be surprising to find that overexpression studies of this type would result in false positives, based on the high expression of the transfected-expressed enzyme and the probable relative promiscuity of the recognition sequences in the mRNA. It should be relatively straightforward to determine mRNA stability in the original cultured HSC by labeling experiments or by using actinomycin D, perhaps supplemented by assays of direct binding (if this can be done without cleaving the mRNA).

In summary, my view is that the authors have not done a convincing job of identifying what they propose as the direct targets of Regnase-1, whose accumulation results in the hematopoietic phenotype they describe. Depending on some of the answers to the questions above, the authors may already have the data for the overall changes in abundance of various transcripts in the control and KO cells, but I think they need to do a much more detailed job of presenting those data as well as confirming the proposed changes in mRNA stability by other means.

Reviewer #3 (Remarks to the Author):

This manuscript by Kidoya et al reports the function of Regnase-1 in regulation of hematopoietic stem cells and hematopoietic pathology. There are a number of novel observations but many conclusions are not convincing and require further development. The stated mechanism that Gata2 and Tal1 hyperactivation mediates the Regnase-1KO HSC self-renewal and leukemia is not well supported. Major concerns:

1. The authors states that Regnase 1 selectively affects self-renewal of LT-HSC but the data to support the selectivity is not strong. First, Fig. 1d c-Kit⁺ cells are not all HSC. Second, Fig.3a and b shows that both LT-HSC and ST-HSC cell cycle is significantly affected. The authors did not show whether Regnase 1 regulate cell cycle for MPP or other later progenitor cells? In addition, the cell number (not just frequency) for LT-HSC, ST-HSC at different cell cycle phase should be shown. Furthermore, the cell number shown in Fig. 3L and mRNA expression shown in Fig. 3m were LSK cells (not LT-HSC). In fact, the authors data showed that there was no difference in the number of CD150⁺CD48⁻ LSK (LT-HSC as defined in Oguro 2013, reference 23) but the number of CD150⁺CD48⁺ LSK (MPP cells as defined in Oguro 2013, reference 23) was significantly increased. The authors observed increase number fo CD34⁻ LSK. Can the authors comment on whether Regnase 1 regulate CD34 expression? It is difficult to reach the conclusion that Regnase 1 has a selective role in the LT-HSC compartment based on the data provided.
2. The conclusion that Regnase 1 does not affect differentiation was also not well supported. Fig. 4e show changes in frequency of lymphoid and myeloid cells. The cell number and the age of mice needs to be shown. If these were 8 weeks old mice, how early could the defects be seen? If early changes in lineage composition were seen, then differentiation could be affected in Reg-1 KO mice.
3. The authors stated that apoptosis of LT-HSC and ST-HSC was not affected in Reg-1 KO, however, Fig. 3i indicate significant ($p < 0.005$) difference.
4. It is not clear what was the criteria for classifying the pathology seen in Reg-1 KO mice as AML-M4. More detailed characterizations need to be provided. Also, is the disease transplantable?
5. It is not clear why the authors switched to Tie2Cre for the RNA-seq experiment. As Tie2Cre is also expressed in endothelial cells, they are not hematopoietic specific as the authors stated. It would be more logical to examine LT-HSC by RNA-seq using the VavCre mice where the hematological phenotypes were observed. The definition of "similarities" between Reg-1KO HSC and AML needs to be described. Also, it is not clear why the authors compared the gene expression of Reg-1KO HSC with human BCR-ABL⁺ AML. If the authors think that the disease seen in Reg-1 KO mice resembles AML-M4, then it would make sense to compare to AML-M4 patients or patients with reduced REG-1 expression. Can the authors show that Regnase-1 expression itself associates with prognosis?
6. The authors show that Regnase-1 protein levels in human AML cell lines were much lowered than healthy PBMC. Does the author know that Regnase-1 is also expressed higher in human HSC compared to more mature populations? The conclusion that Regnase-1 deletion in HSC leads to gene expression signature that is similar to AML HSC is not well supported.
7. The authors claim that Regnase-1 regulates Gata2 and Tal1 mRNA stability. To support this claim, Northern blots showing Gata2 and Tal1 mRNA degradation need to be shown.
8. The authors claim that Gata2 and Tal1 hyperactivation mediates the HSC self-renewal regulated by

Regnase-1. The effects of Gata2 and Tal1 knock-down on Reg-1 KO LT-HSC cell cycle need to be shown to support this claim. It is, however, difficult to ensure that LT-HSC is maintained following the cytokine stimulation and retroviral transduction. Therefore, the effects in proliferation seen is not necessarily effects in LT-HSC. In this context, understanding whether Regnase 1 selectively affects self-renewal of LT-HSC or also affects MPP and later progenitor (comment 1) is important to interpret these results.

Response to Reviewers' comments:

Reviewer #1 (Remarks to the Author):

In this study, the authors found that Regnase-1 is highly expressed in adult hematopoietic stem cells. Regnase-1 knockout in hematopoietic cells results in expansion of hematopoietic progenitors and reduction of quiescent stem cells specifically in bone marrow. Transplantation analyses show that Regnase-1 deficiency impairs repopulation potential of hematopoietic stem cells. These results indicate that Regnase-1 is required for stem cell functions. Cell cycle is activated in Regnase-1-deficient stem cells. Regnase-1 homozygous and heterozygous knockout mice exhibit splenomegaly and lymphadenopathy. Lymphoid cells are decreased and myeloid cells are increased in the Regnase-1 homozygous and heterozygous knockout mice, suggesting that Regnase-1 is indispensable for normal hematopoiesis. Blasts and cells with abnormal morphology were observed in the peripheral blood and bone marrows of Regnase-1-deficient mice. Gene expression profiles of Regnase-1 are similar to those of human hematopoietic stem cells in leukemia patients but not healthy people. Based on these results, the authors insist that Regnase-1 deficiency causes leukemia development. Finally, the authors performed analysis of RNA-seq database and identified Gata2 and Tal1 mRNAs as direct targets of Regnase-1, which account for hematopoietic phenotypes of Regnase-1-deficient mice.

This study provides an interesting insight according to Regnase-1 function in hematopoiesis, especially hematopoietic stem cells. Most of the experiments are carefully done. However, this reviewer feels that the present data is not sufficient to support the authors' conclusion. To support the authors' conclusions, there are several issues that should be addressed:

Response to Reviewer 1

The authors thank the reviewer 1 for his/her review comments and valuable suggestions for improving our manuscript. Please see our responses to your comments one by one below.

Comment 1-1:

1. In Figure 1, the authors analyzed Regnase-1 expression levels of stem cells and progenitors. The authors should examine Regnase-1 expression levels in lineage-committed cells (e.g. B cells, T cells, myeloid, erythroid and megakaryocyte lineages), because the authors showed lineage-skewing in Regnase-1-deficient mice in Figure 4.

Our response to Comment 1-1:

According to the suggestion, we measured the expression of Regnase-1 mRNA in lineage-committed cells including B220⁺ B-cells, CD3⁺ T-cells, CD11b⁺ myeloid-cells, Lin-Sca1⁻cKit⁺CD34⁻CD16/32⁻ erythroid and megakaryocyte lineage cells and other progenitor cells. Results showed that Regnase-1 is more strongly expressed in CD34-negative immature hematopoietic stem cells than in the other cell types. However, low Regnase-1 expression was also observed in lineage-committed cells (B-cells, T-cells and myeloid-cells). Therefore, we cannot exclude the possibility that lineage-skewing of Regnase-1-KO mice resulted from Regnase-1 deficiency in

lineage-committed cells. We added these data in Supplementary Figure 1a and added comments in the Results section (lines 15-18, page 6).

Comment 1-2:

2. In Figure 4, the authors show increased myelopoiesis and decrease in lymphopoiesis in *Regnase-1*-deficient mice. Are they consequences of *Regnase-1* deficiency in HSC (not lineage-committed cells)? As the authors mentioned in Discussion, *Regnase-1* has cell type-specific function. The authors should discuss this issue.

Our response to Comment 1-2:

As suggested, Cre-mediated *Regnase-1* deletion affects not only HSCs but also lineage-committed cells in *Vav1-Cre; Regnase-1^{flox/flox}* mice. As mentioned in response to comment 1, *Regnase-1* expression is also observed in lineage-committed cells, so it is possible that the changes in myelopoiesis and lymphopoiesis observed in *Regnase-1*-KO mice are caused by deletion of *Regnase-1* in lineage-committed cells. We have considered these possibilities in the Discussion section (lines 14-17, page 23).

Comment 1-3:

3. Furthermore, this reviewer wonders whether *Regnase-1* knockout affects erythropoiesis and megakaryopoiesis. Splenomegaly were often observed when abnormal erythropoiesis occurred. The authors should examine erythroid cells and megakaryocytes in the *Regnase-1*-deficient mice.

Our response to Comment 1-3:

Thank you for your constructive criticisms. We examined erythropoiesis in *Regnase-1*-KO mice by counting the number of red blood cells and platelets, evaluating the fraction of megakaryocyte-erythroid progenitor cells (MEP) and erythroid subsets by flow cytometry. As shown in Supplementary Figures 2a and b, red blood cell and platelet counts in peripheral blood and hemoglobin concentrations are significantly decreased in *Regnase-1* KO mice. On the other hand, MEPs and erythroid subset populations are slightly but not significantly changed (Supplementary Figure 2c, d). We also analyzed the megakaryocytes in the bone marrow of *Regnase-1* KO mice by immunohistochemical staining using anti-CD41 antibody (Supplementary Figure 2e). There were no marked differences in the number of CD41⁺ megakaryocytes in *Regnase-1* KO mice. From these results, we consider that *Regnase-1* deficiency has a slight effect on terminal differentiation of erythrocytes, and we have added these results to the Results (lines 15-18, page 13).

Comment 1-4:

4. In Figure 5, the authors insist that *Regnase-1*-deficient mice develop leukemia. How was absolute number (concentration) of blasts in peripheral blood? Is this leukemia transplantable?

Our response to Comment 1-4:

In accordance with this suggestion, we examined the frequency of blasts in peripheral blood. As shown in Figure 5c, numerous blasts ($10.8\% \pm 6.3\%$) and abnormally differentiated cells were observed in Regnase-1 KO mice.

To confirm that Regnase-1 deficiency-induced leukemia cells are transplantable, we transferred bone marrow mononuclear cells from Regnase-1-KO mice to C57BL/6-Ly-5.1 congenic mice. 3 months after transplantation, leukemia-like symptoms such as abnormal proliferation of CD34⁺LSK cells and splenomegaly were confirmed (Supplementary Figure 5). Based on this study, we concluded that leukemia caused by Regnase1-deficiency is transplantable and have added this to the text (lines 16-19, page 15).

Comment 1-5:

5. The authors identified Gata2 and Tal1 as responsible genes. Are the cell cycle-related genes p21 and p57 shown in Figure 3 direct targets of GATA2 or TAL1?

Our response to Comment 1-5:

We investigated the direct association of Gata2/Tal1 on p21/p57 gene expression by chromatin immunoprecipitation assays. Although Regnase-1 expression induces the upregulation of p21 and p53 mRNA in THP-1 leukemic cells, direct binding of Gata2 or Tal1 to p21 or p57 promoter regions was not observed (Supplementary Figure 6). Thus, we conclude that Gata2/Tal1 indirectly regulate p21 and p53 mRNA expression. We now describe these issues in the Discussion (lines 5-8, page 25).

Reviewer #2 (Remarks to the Author):

Kidoya et al # NCOMMS-17-23012-T - Regnase-1-mediated post-transcriptional regulation is essential for hematopoietic stem cell homeostasis

These authors address a potential role for Regnase-1, a CCCH zinc finger RNA binding protein and RNase, in the regulation of hematopoiesis. Their major finding is that “Regnase-1 regulates self-renewal of HSC through modulating the stability of Gata2 and Tal1 mRNA.” They also found that “dysfunction of Regnase-1 leads to rapid onset of AML”, and concluded that “Regnase-1-mediated post-transcriptional regulation is required for HSC maintenance and suggest that it represents a novel type of leukemia tumor suppressor.”

I will leave it to other reviewers to focus on the validity of the findings relating to the behavior of hematopoietic stem cells in the mutant mice, as well as the importance of these findings to the overall field of hematopoiesis regulation. My comments will focus on the RNA-Seq data and the conclusions regarding direct effects on Gata2 and Tal1 mRNA stability.

Despite going back and forth between the text, the methods section, and the figure legends, I was not able to figure out several aspects of the RNA-Seq data. It is possible that these pieces of information were hidden in other places, but I was not able to find them. I will list here some of my major problems with the information presented, but an

overall comment is that it would be very helpful for the reviewer to have all the relevant information in one place, for example, in the methods or results section. Specific issues include, but are not limited to:

Response to Reviewer 2

The authors thank the reviewer 2 for his/her review comments and valuable suggestions for improving our manuscript. As the reviewers pointed out, there were several ambiguous descriptions in terms of the RNA-seq data analysis. According to the suggestion, we described all the relevant information to RNA-seq analysis in methods section. We have described our responses to your comments one by one as below. Moreover, in order to show in detail the “identification and variation of target genes of Regnase-1”, we divided original Figure 6 into two (Figure 6 and Figure 7) and added data.

Comment 2-1:

1. How were the HSC treated or cultured after the FACS separation? To my knowledge, we have no information on that.

Our response to Comment 2-1:

Thank you for the comment. In almost all experiments, except for the knock-down experiment of Figure 7, HSCs were used for analysis directly after FACS separation. We inserted an explanation of methods for using HSCs in the Methods section (lines 15-16, page 31).

Comment 2-2:

2. How many biological replicates were used to generate the data in Fig. 5f and Fig. 6, showing the overall differential expression, i.e., how many mice were used to prepare how many independent cultures of HSC used in this experiment?

Our response to Comment 2-2:

We used HSCs collected from one individual mouse of each genotype in Figures. 5f-i and Figure. 6. HSCs separated and sorted by flow cytometry were directly analyzed without culture. We inserted an explanation of the sample preparation method for RNA-seq analysis to the Figure legends.

Comment 2-3:

3. Regardless of the answers to 2 above, was the RNA-Seq experiment performed more than once?

Our response to Comment 2-3:

Thank you for your question. We performed two independent RNA-Seq experiments and similar results were

obtained. We added this information to the Figure legends.

Comment 2-4:

4. What was the statistical method used to determine statistical significance of the differences?

Our response to Comment 2-4:

We detected 801 up-regulated genes and 1179 down-regulated genes in the $Reg1^{\Delta/\Delta}$ according to the following criteria: log₂ fold change of the average expression value in $Reg1^{\Delta/\Delta}$ mice (case) compared with that in $Reg1^{flox/flox}$ mice (control) was larger than 2 or smaller than 1/2, and the average expression value in the case was larger than the 90th percentile of the case. We added the data in Supplementary table 4 and added this explanation into the Methods section (line 16, page 38 - line 5, page 39). As described in the Methods section (lines 5-11, page 39), GSEA (Figures 5g, h) was performed with FDR < 0.1 and p-value < 0.05 as significant and survival analysis (Figure 5i) was performed using the Kaplan-Meier method and the p-value was calculated using the log-rank test. Because there is no biological replicate in IGV-Sashimi analysis (Figure 6a), we did not perform statistical analysis in that experiment.

Comment 2-5:

5. Were the transcript data limited by an abundance cutoff?

Our response to Comment 2-5:

We had described criteria and cutoff for selection of candidate genes (Figure 6b) and for identification of Regnase-1 (Figure 1a) in the Methods section (lines 1-13, page 38) and (lines 6-17, page 37). There is no cutoff for GSEA (Figures 5f, g).

Comment 2-6:

6. Although the researchers took their gene sets through many screens before settling on the handful that they analyzed further as potential direct targets of Regnase-1, I think the reader should be given easy access to the actual gene lists, showing fold changes, an indication of abundance, and an indication of the significance of the fold changes in the original normalized RNA-Seq data. Perhaps a supplemental Excel file could be used to address most or all of these questions, as well as certain fundamental questions like the relative abundance of housekeeping transcripts, the abundance of the knocked out gene (reflecting cell purity), etc. In my view, all of this information is necessary to help the reader determine the validity of the authors' conclusions on this topic.

Our response to Comment 2-6:

We thank the reviewer for this constructive criticism. In accordance with the reviewer's suggestion, we now add

actual gene lists of original normalized RNA-Seq data including fold-changes and an indication of abundance (Supplementary Table 3). Significance of the fold-changes and abundance cutoff value described in the Methods section is also mentioned in this list.

Comment 2-7:

7. The authors then moved to their analysis of potential target transcripts. Rather than starting from the total collection of up-regulated transcripts, which most likely included many transcripts regulated by Regnase-1 either directly or secondarily, the authors moved to their exon gradient analysis, and limited the data still further by screening for up-regulated transcripts that are known to be involved in hematopoiesis. They thereby eliminated from the analysis many potential target transcripts that might well be involved in cellular function because they did not meet those screening criteria.

Our response to Comment 2-7:

Regarding this comment, we are not sure exactly what your question is and are therefore wondering if the response below is adequate or not; however, this is as follows. Regnase-1 binds to stem-loop element in the 3' UTR of target mRNAs and promotes degradation via its intrinsic RNase activity. While there are several studies on the binding sequences of Regnase-1, information about its interaction with target RNAs is still limited (Behrens G, et al., *Nucleic Acids Res.* 2018., Mino T, et al., *Cell.* 2015). It appears that Regnase-1 degrades various target mRNAs (Jeltsch KM and Heissmeyer V. *Curr Opin Immunol.* 2016); thus it seems difficult to identify all of the target genes. Because the results of RNA-Seq analysis show that expression of numerous genes is greatly affected by Regnase-1 deficiency, we focused on transcription factors that influence the expression of a wide range of genes. We have added this information and explanation to the Results section (lines 8-11 page 17).

Comment 2-8:

8. For Fig. 6c, I assume that these data were from real-time QPCR, but it is not clear that this is the case from the Results section.

Our response to Comment 2-8:

As pointed out, data of Figure 6c were obtained by real-time QPCR analysis. We have changed “quantitative PCR” to “real-time QPCR” (lines 18 page 17).

Comment 2-9:

9. It would be very helpful in the section on target transcript analysis to review the binding and/or cleavage sites for this protein, and the evidence to support these.

Our response to Comment 2-9:

We appreciate the reviewer's advice. As mentioned in "Our response to Comment 2-7", Regnase-1 binds to the 3' UTR of target mRNAs and promotes degradation. However, reports about the Regnase-1 binding sequences and mechanism of cleavage are still limited (Behrens G, et al., *Nucleic Acids Res.* 2018). Because the molecular function of Regnase-1 has not been clearly elucidated, we have added a complementary explanation. Cleavage sites of target mRNA are not clear, but our data from sashimi IGV analysis show a gradient in 3' and 5' mRNA amounts of Gata2 and Tal1 in HSC from control mice relative to Regnase-1-deficient mice. These results indicate that degradation of Gata2 and Tal1 was induced in the presence of Regnase-1. To make this clearer to understand, Bmi1 which is a transcriptional regulator important for HSC proliferation and differentiation, was added as an example of an undegraded molecule in Figure 6a.

Comment 2-10:

10. I don't think the co-transfection assays with luciferase readouts are adequate demonstrations that the mRNAs studied are direct targets in vivo. It would not be surprising to find that overexpression studies of this type would result in false positives, based on the high expression of the transfected-expressed enzyme and the probable relative promiscuity of the recognition sequences in the mRNA. It should be relatively straightforward to determine mRNA stability in the original cultured HSC by labeling experiments or by using actinomycin D, perhaps supplemented by assays of direct binding (if this can be done without cleaving the mRNA).

Our response to Comment 2-10:

As indicated by the reviewer, overexpression studies often result in abnormal conditions for the cell and thereby cause artefactual effects. To exclude this, we performed actinomycin D blockade experiments using primary cultured Regnase-1-deficient HSC, because this design was used in a previous study of Regnase-1 (Uehata T, et al., *Cell.* 2013). Marked Gata2 and Tal1 mRNA degradation was observed in HSCs from control mice, but this effect was abrogated in HSCs from Regnase-1-deficient mice. We have added these results to Figure 6f (lines 12-14, page 18).

In summary, my view is that the authors have not done a convincing job of identifying what they propose as the direct targets of Regnase-1, whose accumulation results in the hematopoietic phenotype they describe. Depending on some of the answers to the questions above, the authors may already have the data for the overall changes in abundance of various transcripts in in the control and KO cells, but I think they need to do a much more detailed job of presenting those data as well as confirming the proposed changes in mRNA stability by other means.

Reviewer #3 (Remarks to the Author):

This manuscript by Kidoya et al reports the function of Renase-1 in regulation of hematopoietic stem cells and

hematopoietic pathology. There are a number of novel observations but many conclusions are not convincing and require further development. The stated mechanism that Gata2 and Tal1 hyperactivation mediates the Regnase-1KO HSC self-renewal and leukemia is not well supported.

Response to Reviewer 3

Thank you very much for your review comments and valuable suggestions to improve our manuscript. In response, we have added new data and further discussion. The definition of HSC was ambiguous in the manuscript, thus we described it more exactly by rewriting the terms (e.g., “LT-HSC” into “CD34⁻ HSC”). It was not clear whether AML-like phenotype of Vav1-Cre; Regnase-1^{flox/flox} mice was caused by Regnase-1 deficiency or Gata2 and Tal1 hyperactivation in HSC. In accordance with reviewer’s suggestions, we examined the influence of blood cell differentiation or hematocyte proliferation for leukemia symptom. Please see our responses to your comments one by one below.

Comment 3-1:

Major concerns:

1. The authors states that Regnase-1 selectively affects self-renewal of LT-HSC but the data to support the selectivity is not strong. First, Fig. 1d c-Kit⁺ cells are not all HSC. Second, Fig.3a and b shows that both LT-HSC and ST-HSC cell cycle is significantly affected. The authors did not show whether Regnase-1 regulate cell cycle for MPP or other later progenitor cells? In addition, the cell number (not just frequency) for LT-HSC, ST-HSC at different cell cycle phase should be shown. Furthermore, the cell number shown in Fig. 3L and mRNA expression shown in Fig. 3m were LSK cells (not LT-HSC). In fact, the authors data showed that there was no difference in the number of CD150⁺CD48⁻ LSK (LT-HSC as defined in Oguro 2013, reference 23) but the number of CD150⁺CD48⁺ LSK (MPP cells as defined in Oguro 2013, reference 23) was significantly increased. The authors observed increase number of CD34⁻ LSK. Can the authors comment on whether Regnase-1 regulate CD34 expression? It is difficult to reach the conclusion that Regnase-1 has a selective role in the LT-HSC compartment based on the data provided.

Our response to Comment 3-1:

1. We thank the reviewer for this constructive criticism. The definition of HSC subsets was imprecise in many parts of the manuscript, and for this reason we have now rewritten all descriptions of HSCs more precisely (“LT-HSC” is now “CD34⁻ HSC”, “ST-HSC” is now “CD34⁺HSC”). As indicated by the reviewer, c-kit⁺ cells in Figure 1d are not always HSCs, and we rewrote this to read “c-kit⁺ cells including hematopoietic progenitor cells”. In the same way, we corrected Figure 3l, m. Regarding the cell cycle assay, we analyzed the cell cycle of MPP and Lineage⁺ cells, and display the cell number in each cell cycle phase including LT-HSC (CD34⁻ HSC) and ST-HSC (CD34⁺ HSC) as requested by the reviewer. In both CD34⁻ HSCs and CD34⁺ HSCs, the number of cells in G0/G1 phase, S phase, and G2/M phase markedly increased, associated with enhancement of cell proliferation. In particular, significant increases in cell numbers were observed in S phase CD34⁻ HSCs. Marked differences in the number of cells at each

phase of the cell cycle were not seen in MPP, and were only slightly increased in the G0/G1 phase. These results indicate that Vav1-Cre-mediated deletion of Regnase-1 mainly affects the cell cycle of CD34⁻ HSC (LT-HSC) and CD34⁺ HSC (ST-HSC) fractions. We have added these results to Figure 3a and Supplementary Figure 1b (lines 13-18, page 11), and indicated that the phenotype of Vav1-Cre; Regnase-1^{flx/flx} mice is not caused only by abnormalities of CD34⁻ HSCs (LT-HSCs) in the Discussion (lines 14-17, page 23).

In accordance with the reviewer's suggestion, we investigated whether Regnase-1 degrades CD34 mRNA in CD34⁻ HSCs (LT-HSCs). CD34⁻ HSCs collected from control and Regnase1-KO mice were cultured under conditions where RNA synthesis was inhibited by actinomycin D, and the degradation of CD34 mRNA was examined. Gata2 and Tal1 mRNA degradation was suppressed in CD34⁻ HSCs from Regnase-1 KO mice, but CD34 mRNA degradation was not significantly different between control and Regnase-1 KO CD34⁻ HSC as shown in Figure 6f (lines 12-14, page18). From this result, we considered that CD34 could be used as a marker not affected by Regnase-1.

Comment 3-2:

2. The conclusion that Regnase-1 does not affect differentiation was also not well supported. Fig. 4e show changes in frequency of lymphoid and myeloid cells. The cell number and the age of mice needs to be shown. If these were 8 weeks old mice, how early could the defects be seen? If early changes in lineage composition were seen, then differentiation could be affected in Reg-1 KO mice.

Our response to Comment 3-2:

Thank you for the important suggestions. We used 8 week-old mice for lymphoid and myeloid cell analysis in Figure 4e. We added this information to the Figure legend and replaced the graph showing the cell number. In addition, we examined differentiated cells in peripheral blood and bone marrow from 2 week-old mice. There were abnormal HSCs in 2 week-old mice (Figure 2a), and moderate abnormalities were observed in the number of differentiated cells of Regnase-1 KO mice (Supplementary Figure 3, lines 2-4, page 14). The proportion of GM cells in the colony assay as shown in Figure 2e was slightly changed in CD34⁻ HSC from Regnase-1 KO mice, and an influence on differentiation has also been observed in GO analysis (Figure 2j). Thus, we considered that Regnase-1 may affect hematopoietic differentiation. We have added this information to the Discussion (lines 9-11, page 21).

Comment 3-3:

3. The authors stated that apoptosis of LT-HSC and ST-HSC was not affected in Reg-1 KO, however, Fig. 3i indicate significant ($p < 0.005$) difference.

Our response to Comment 3-3:

We thank the reviewer for pointing out our mistake. The notation of Figure 3i was an error and we have corrected

this in the revised manuscript.

Comment 3-4:

4. It is not clear what was the criteria for classifying the pathology seen in Reg-1 KO mice as AML-M4. More detailed characterizations need to be provided. Also, is the disease transplantable?

Our response to Comment 3-4:

In accordance with the reviewer's suggestion we described the details of the pathological examination in Results (lines 2-8, page 15). In the peripheral blood from Regnase-1 KO mice, we observed 1 to 14% of lymphoblasts and 16 to 22% of atypical lymphocytes exhibiting abnormal nuclear morphology, 2 to 4% of mature myelocyte and metamyelocyte and 1 to 16% of atypical cells including stab cells in neutrophils. In addition, 3 to 8% of monoblasts and 6 to 20% of atypical mononuclear cells were present, and proerythroblast-polychromatophilic erythroblasts were observed. Furthermore, 1% of myeloblasts were found. Therefore, we diagnosed AML.

To investigate whether the AML was transplantable, we transferred mononuclear cells (MNCs) from bone marrow of Regnase1-KO mice into C57BL/6-Ly-5.1 recipient mice. As shown in Supplementary Figure 5, abnormal proliferation of Ly 5.1-negative donor-derived LSK cells and CD34⁺ HSCs was observed in bone marrow of these transplanted mice. In addition, they showed splenomegaly similar to Regnase-1 KO mice. These results indicate that Regnase-1 KO-derived leukemia cells are transplantable. We now describe this in the Results section (lines 16-19, page 15).

Comment 3-5:

5. It is not clear why the authors switched to Tie2Cre for the RNA-seq experiment. As Tie2Cre is also expressed in endothelial cells, they are not hematopoietic specific as the authors stated. It would be more logical to examine LT-HSC by RNA-seq using the VavCre mice where the hematological phenotypes were observed. The definition of "similarities" between Reg-1KO HSC and AML needs to be described. Also, it is not clear why the authors compared the gene expression of Reg-1KO HSC with human BCR-ABL⁺ AML. If the authors think that the disease seen in Reg-1 KO mice resembles AML-M4, then it would make sense to compare to AML-M4 patients or patients with reduced REG-1 expression. Can the authors show that Regnase-1 expression itself associates with prognosis?

Our response to Comment 3-5:

As indicated by the reviewer, Tie2Cre causes recombination in vascular endothelial cells in addition to blood cells. We performed RNA-seq analysis by using CD34⁺ HSCs from BM of Vav-Cre Reg1^{fllox/fllox} mice and obtained similar results (please see the additional figure for the reviewer). However, these RNA-seq data seem technically noisy, and we were unable to acquire precisely the same results especially in IGV-Sashimi plot analyses (Figure 7a). Technical noise of low-input RNA-seq analysis is reported to affect downstream analyses for detecting subtle

biological differences of gene expression in small amounts of cells (Sheng K, et al. Nat Methods. 2017., Jia C, et al. Nucleic Acids Res. 2017). Therefore, we have omitted the data from Vav-Cre Reg1^{fllox/fllox} mice from the manuscript, and retained the previous results from Tie2Cre Reg1^{fllox/fllox} mice. We anticipate that the reviewer will appreciate this technical limitation. Regarding the subtype of AML, to properly respond to comment 3-4, we repeated the experiments using more mice. As a result, a wide range of AML-like cells representing not only specifically AML-M4 was confirmed. Therefore, we have corrected the description of the AML-M4 phenotype in the text and have performed GSEA analysis with human AML-M1 or -M3 cells (Figure 5g). The “similarities of GSEA” means that sample genes sets show statistically significant concordant differences with *a priori*-defined sets of genes which are known to have an association with disease phenotypes. We have added this information to the Results section (lines 5-8, page 16). Correlations between Reg1 mRNA expression and AML prognosis were analyzed by Prognoscan (Hideaki Mizuno, et al., BMC Med Genomics. 2009), but there was no significant difference (corrected p-value and Cox p-value shown in Supplementary Table 1, lines 11-14, page 16). As mentioned in the Discussion section (line 16, page 25 - line 1, page 26), the reason is that Regnase-1 expression is regulated by protein modification such as phosphorylation and ubiquitination.

Comment 3-6:

6. The authors show that Regnase-1 protein levels in human AML cell lines were much lowered than healthy PBMC. Does the author know that Regnase-1 is also expressed higher in human HSC compared to more mature populations? The conclusion that Regnase-1 deletion in HSC leads to gene expression signature that is similar to AML HSC is not well supported.

Our response to Comment 3-6:

We thank you for your constructive criticisms. We agree with the comment that analysis of Regnase-1 expression in human HSC was necessary in the context of relevance for human AML. We evaluated protein expression of Regnase-1 in human BM-derived CD34⁺ HSCs, which are commercially available. As shown in Supplementary Figure 4d and the Results section (lines 16-18, page 16), high expression of Regnase-1 was observed in human HSC relative to PBMC.

Comment 3-7:

7. The authors claim that Regnase-1 regulates Gata2 and Tal1 mRNA stability. To support this claim, Northern blots showing Gata2 and Tal1 mRNA degradation need to be shown.

Our response to Comment 3-7:

As suggested by the reviewer, it is important to show the degradation of Gata2 and Tal1 mRNA in the presence of Regnase-1. In accordance with this suggestion, we performed Northern blotting and added these data to Figure 6e and the Results (lines 8-14, page 18). Because it is a challenge to prepare sufficient mouse HSCs for these assays,

we analyzed a THP 1 leukemia cell line overexpressing Regnase-1. As shown in Supplementary Figure 6a, we confirmed that Regnase-1 overexpression decreases Gata2 and Tal1 mRNA expression. Northern blotting revealed that overexpression of Regnase-1 causes degradation of Gata2 and Tal1 mRNA (Figure 6e). Moreover, we analyzed mRNA degradation using actinomycin D, in order to respond to Reviewers' comments 2-3 and 3-1, and confirmed that half-lives of Gata2 and Tal1 mRNA were significantly extended in Regnase-1-deficient CD34⁻ HSCs compared with CD34⁻ HSCs from control mice (Figure 6f).

Comment 3-8:

8. The authors claim that Gata2 and Tal1 hyperactivation mediates the HSC self-renewal regulated by Regnase-1. The effects of Gata2 and Tal1 knock-down on Reg-1 KO LT-HSC cell cycle need to be shown to support this claim. It is, however, difficult to ensure that LT-HSC is maintained following the cytokine stimulation and retroviral transduction. Therefore, the effects in proliferation seen is not necessarily effects in LT-HSC. In this context, understanding whether Regnase-1 selectively affects self-renewal of LT-HSC or also affects MPP and later progenitor (comment 1) is important to interpret these results.

Our response to Comment 3-8:

To determine whether Gata2 and Tal1 hyperactivation are involved in the cell cycle abnormalities seen in Regnase-1-deficient HSCs, Gata2 and Tal1 expression by CD34⁻ HSCs (LT-HSCs) from Regnase-1 KO BM was downregulated by shRNA transfection and then those cells were transplanted into C57BL/6-Ly-5.1 congenic mice. As in the analysis of Figure 7 f-h, 2 months after transplantation, we confirmed that abnormal proliferation was suppressed in transplanted HSCs; we then analyzed the cell cycle of CD34⁻ HSCs and confirmed that cell cycle abnormalities in CD34⁻ HSCs due to Regnase-1-deficiency was partially recovered by knockdown of Gata2 and Tal1 expression. We have added these results to Figure 7h and the Results section (lines 14-18, page 19). As described in response to Comment 3-1, cell cycle abnormalities due to Regnase-1 deletion is mainly observed in CD34⁻ HSCs and CD34⁺ HSCs; thus we considered that hyperactivation of Gata2 and Tal1 in immature HSCs may be involved in the development of AML-like symptoms in Regnase-1 KO mice.

Reviewers' comments:

Reviewer #1 (Remarks to the Author):

The authors well answered my concerns. In the rebuttal, the authors found that Regnase-1 is expressed in differentiating cells, implying that Regnase-1 may play a role in the differentiation of lineage-committed cells. This reviewer expects that the authors examine this issue in their next study.

Reviewer #2 (Remarks to the Author):

In my view, this revised paper did not correct many of the fundamental problems that were present in the original. For example, focusing on the differential gene expression aspects of the paper, they clarified that they were using a single mouse per genotype, and a single biological replicate, for their analysis pictured in Fig. 5e. They mentioned in the figure legend for Fig. 5e that there were two independent experiments, but they did not clarify whether Fig. 5E represented a single experiment or was an average of the two. In any case, in my opinion, this number of biological replicates is considerably fewer than most journals and reviewers would accept for adequately controlled, statistically validated differential expression analysis. As best I can tell, the authors provided a bit more explanation for this but did not address the fundamental problem, i.e., too few biological samples.

Another fundamental problem that was addressed in the original review was that most of the experiments were performed with mice expressing one particular Cre, whereas the differential expression analysis used a completely different Cre. In the authors' response, they said that they obtained "similar results" using the Vav-cre mice, but that the data seemed "technically noisy, and we were unable to acquire precisely the same results". I don't think this is an adequate explanation.

My view is that the differential gene expression aspects of the paper are fundamental to their conclusions, but, based on some of the factors enumerated above, we are unable to rely on these data as presented.

Reviewer #3 (Remarks to the Author):

In general, this manuscript by Kidoya et al has improved but a number of significant concerns remain. Particularly, it is not convincing that the pathology seen in the Reg1 deficient mice are spontaneous AML. It is generally thought that AML pathogenesis require multiple "hits". It seems quite unlikely that there is full malignant transformation in 2-3 months. In addition, CD34-HSC transplant do not seem to develop leukemia-like pathology, this argues against Regnase1 loss is sufficient to cause leukemia. Given the role of Regnase1 in regulating inflammatory genes in immune cells, it is important to consider possible consequences on the observed phenotype not necessarily the consequences of its role in CD34-HSC.

Major issues:

1. If Reg1 loss accelerates CD34-HSC cell cycling as the author show, it is possible that would lead to HSC exhaustion and explain the results shown for transplantation experiment (Fig. 2f,h,i,g). Is there expansion of progenitors at earlier time points? Do the author have engraftment/repopulation data for earlier time points? It would help clarify the effects on progenitor proliferation/differentiation and HSC exhaustion.

2. It is rather confusing to state that Reg1 deficiency leads to bone marrow failure like disease (non-malignant) and promote leukemogenesis (malignant) in what appears to be the same animals and same time point? Was there increase total WBC counts in the diseased mice? The cell numbers in PB shown in Fig 4e, m seem very low and the abnormal cells seen in the PB includes both lymphoid and myeloid cells (Fig. 5c). These observations are not typical for a diagnosis of AML as the authors claimed. The authors now show transplantation of cells from diseased mice but it is not mentioned whether these mice also developed "leukemia". A transplantable disease means that the disease phenotypes, including major pathological features defining the types of leukemia in this case is reproduced in the transplants. Given the data presented here, it is difficult to conclude that the abnormality seen in Reg1KO at 8 weeks is AML. It remains possible that what is seen is some form of MPD or MPN-like condition. The authors further made a strong statement that "results in the spontaneous development of leukemia even in heterozygous mice" (Page 26, line 3). There is very limited characterization of the heterozygous mice to support this claim.

3. Down regulation of Regnase1 in AML cells is not sufficiently supported by just comparing AML cell line to PBMC. Particularly since the author showed that Regnase1 is also expressed in lymphoid cells (and appears to be higher than myeloid cells) (Supplemental Figure 1a), the lower levels seen could be just due to the different cellular composition.

Minor issues:

1. In Fig. 1c, which one is CD34-HSC?

2. In Fig. 2d, are all cells CD34-? Is the expansion of CD48+LSK shown in Fig 2d related to changes cell cycle?

3. Fig. 2g show increased % of CD34- cells in donor derived LSK for Regnase-1KO but there is no mention about absolute cell number. Given that the % of donor derived LSK is greatly reduced, this information alone is not sufficient to conclude that self-renewal capacity of CD34-HSC is increased (page 10). Also, can the authors explain the difference here (almost no CD34+HSC cells vs. increased CD34+HSC cells in Fig 1i) and also no difference in proliferation in Fig. 2e?

4. Fig.3 e,f show significant increase of all cell cycle phases (G0, G1, S/G2M) in Reg1KO CD34-HSC but the text mention reduction of G0 phase (page 12, line 3)? Which is correct? If all phases of cell cycle are increased, how to interpret this?

5. Page 21, line 13 in discussion, should be "promoting" cell cycle progression, not suppressing.

6. Page 21, Line 16-17, please clarify where "Regnase-1 is down-regulated in human AML cells and that its expression is inversely associated with AML prognosis" is shown.

7. Page 25 line 13, not sufficient evidence to suggest that Regnase-1 is the "causative gene for human AML. Line 16, "Regnase-1 might be useful for predicting AML prognosis" contradicts the author the main text that Ragnase 1 expression is not associated with prognosis.

Response to Reviewers' comments:

Reviewer #1 (Remarks to the Author):

The authors well answered my concerns. In the rebuttal, the authors found that Regnase-1 is expressed in differentiating cells, implying that Regnase-1 may play a role in the differentiation of lineage-committed cells. This reviewer expects that the authors examine this issue in their next study.

Reviewer #2 (Remarks to the Author):

Thank you for your valuable comments on our paper which we have amended according to your suggestions.

Comment 2-1:

In my view, this revised paper did not correct many of the fundamental problems that were present in the original. For example, focusing on the differential gene expression aspects of the paper, they clarified that they were using a single mouse per genotype, and a single biological replicate, for their analysis pictured in Fig. 5e. They mentioned in the figure legend for Fig. 5e that there were two independent experiments, but they did not clarify whether Fig. 5E represented a single experiment or was an average of the two. In any case, in my opinion, this number of biological replicates is considerably fewer than most journals and reviewers would accept for adequately controlled, statistically validated differential expression analysis. As best I can tell, the authors provided a bit more explanation for this but did not address the fundamental problem, i.e., too few biological samples.

Our response to Comment 2-1:

Regarding the results of gene expression analysis by RNA-seq, we had performed two independent experiments using a single mouse per genotype, and shown the average of the two. As you pointed out, there are too few biological samples and the results are unreliable. In order to deal with this problem, it is necessary to perform multiple experiments with increased numbers of biological samples. Hence, we performed two independent experiments with $n=3$ per genotype and $n=2$ per genotype. As a result of this approach, we have now obtained data on gene expression with greater reproducibility, and have again averaged them. These new results are very similar to the previous data in the first version of the paper, but we do consider them to be more reliable now. They now replace the previous figure (Figure 2j and 5f-i).

Comment 2-2:

Another fundamental problem that was addressed in the original review was that most of the experiments were performed with mice expressing one particular Cre, whereas the differential expression analysis used a completely different Cre. In the authors' response, they said that they obtained "similar results" using the Vav-cre mice, but that the data seemed "technically noisy, and we were unable to acquire precisely the same results". I don't think this is an

adequate explanation.

Our response to Comment 2-2:

As you pointed out, despite the fact that we had performed all biochemical analyses using Vav1-iCre : Reg1^{fl/fl} mice, bioinformatics analysis of RNA-seq was performed using Tie2-Cre : Reg1^{fl/fl} mice. To improve the reliability of this research by ensuring consistency, we have now performed RNA-seq analysis using Vav1-iCre : Reg1^{fl/fl} mice. The results are similar to the previous data from Tie2-Cre: Reg1^{fl/fl} mice, but not identical. In particular, we could not reproduce the data shown in the previous Figure 6 a,b, and we have therefore deleted this. We searched for candidate genes affected by Regnase expression, not only by informatics analysis but also by biochemical analysis using luciferase-based screening assays. We have now added this information and the data to the Results section (line 17, page 17 – line 9, page 18).

Regarding the reason why the results of gene expression analysis were different between Tie2-Cre : Reg1^{fl/fl} and Vav1-iCre : Reg1^{fl/fl}, it seems that Tie2Cre causes recombination in vascular endothelial cells in addition to hematopoietic cells, whereas Vav1Cre acts only in hematopoietic cells specifically.

Comment 2-3:

My view is that the differential gene expression aspects of the paper are fundamental to their conclusions, but, based on some of the factors enumerated above, we are unable to rely on these data as presented.

Our response to Comment 2-3:

We believe that the quality and reliability of this paper have been improved greatly by using the appropriate number of mice (Vav1-iCre : Reg1^{fl/fl}) so that there are sufficient biological samples and new data included. I'd like to thank you for pointing out the essential issues to improve the paper

Reviewer #3 (Remarks to the Author):

In general, this manuscript by Kidoya et al has improved but a number of significant concerns remain. Particularly, it is not convincing that the pathology seen in the Reg1 deficient mice are spontaneous AML. It is generally thought that AML pathogenesis require multiple “hits”. It seems quite unlikely that there is full malignant transformation in 2-3 months. In addition, CD34-HSC transplant do not seem to develop leukemia-like pathology, this argues against Regnase1 lost is sufficient to cause leukemia. Given the role of Regnase1 in regulating inflammatory genes in immune cells, it is important to consider possible consequences on the observed phenotype not necessarily the consequences of its role in CD34-HSC.

Response to Reviewer 3

Thank you for your valuable comments on our paper which we have amended according to your suggestions.

As you say, AML develops through multiple genetic hits and we agree that it is unlikely that leukemia was caused

solely by Regnase-1 deficiency. Therefore, we have added the following sentence to the paper: “Excessive proliferation of HSPC induced by deficiency of Regnase-1 may provide the possibility to progress to AML by promoting mutagenesis.” in Discussion (lines 18-19, page 21). In addition, we changed the designation from “AML and leukemia” into “AML-like phenotype”, because the phenotype of Regnase-1-deficient mice does not completely match AML, as you point out. If you recommend that we should describe the phenotype as AML from the above results nonetheless, we would be glad to re-write this again.

Moreover, responding to major comment 2, we analyzed peripheral blood from CD34⁺ HSC-transplanted mice and confirmed that they exhibited an AML-like phenotype. Furthermore, in the Discussion, we introduce the possibility that the phenotype of Regnase-1-deficient mice may be caused by an effect on differentiated blood cells.

Major issues:

Comment 3-1:

1. If Reg1 loss accelerates CD34-HSC cell cycling as the author show, it is possible that would lead to HSC exhaustion and explain the results shown for transplantation experiment (Fig. 2f,h,I,g). Is there expansion of progenitors at earlier time points? Do the author have engraftment/repopulation data for earlier time points? It would help clarify the effects on progenitor proliferation/differentiation and HSC exhaustion.

Our response to Comment 3-1:

Thank you for this constructive criticism. In accordance with this suggestion, we have investigated whether transplanted CD34⁺ HSCs differentiated and expanded at the earlier time point of one month after transplantation. By flow cytometric analysis of BM cells from transplanted mice, we observed an increase of CD34⁺ LSK cells and EMPs in the group transplanted with CD34⁺ HSCs from Regnase-1-deficient mice (Supplementary Figure 1d). From this result, we suggest that HSC exhaustion may be caused by accelerated cell cycling and differentiation of HSCs. We have now added these results (Fig. 2f-i) and described them in the Results section (line 18, page 10 - line 4, page 11).

Comment 3-2:

2. It is rather confusing to state that Reg1 deficiency leads to bone marrow failure like disease (non-malignant) and promote leukemogenesis (malignant) in what appears to be the same animals and same time point? Was there increase total WBC counts in the diseased mice? The cell numbers in PB shown in Fig 4e, m seem very low and the abnormal cells seen in the PB includes both lymphoid and myeloid cells (Fig. 5c). These observations are not typical for a diagnosis of AML as the authors claimed. The authors now show transplantation of cells from diseased mice but it is not mentioned whether these mice also developed “leukemia”. A transplantable disease means that the disease phenotypes, including major pathological features defining the types of leukemia in this case is reproduced in the transplants. Given the data presented here, it is difficult to conclude that the abnormality seen in Reg1KO at 8 weeks is AML. It remains possible that what is seen is some form of MPD or MPN-like condition. The authors further made a strong statement that “results in the spontaneous development of leukemia even in heterozygous mice” (Page 26,

line 3). There is very limited characterization of the heterozygous mice to support this claim.

Our response to Comment 3-2:

Thank you for these important suggestions. As you pointed out, the exact phenotype of Regnase-1-deficient mice needs to be described. Because we could not clarify whether “bone marrow failure” is really appropriate to describe the symptoms of Regnase-1-deficient mice, we have deleted this phrase “Bone marrow failure”.

In terms of WBC counts, we have analyzed the number of WBC in Regnase-1-KO mice and have observed increases in these cells (line 15, page 15, Figure 5d). Moreover, to test whether the Regnase-1 deficiency-induced leukemic cells are transplantable, we transferred CD34⁺ LSK cells from Regnase-1-KO mouse bone marrow to C57BL/6-Ly-5.1 congenic mice. Three months after transplantation, we analyzed recipient peripheral blood and found that abnormal blood cells were present, but no blasts (lines 7-8, page 16, Supplementary Figure 5d). Furthermore, in mice heterozygous for Regnase-1, pathological examination of the peripheral blood of such 1-year old heterozygotes also revealed an AML-like phenotype, as in Regnase-1-KO mice (lines 16-17, page 15, Supplementary Figure 4c). Nonetheless, we agree that these results of peripheral blood analyses may not to be sufficient to diagnose AML, as you have indicated. Therefore, we now refer throughout the paper to Regnase-1 deficiency causing “aberrant differentiation” instead of “AML and leukemia” (lines 10-11, page 16).

Comment 3-3:

3. Down regulation of Regnase1 in AML cells is not sufficiently supported by just comparing AML cell line to PBMC. Particularly since the author showed that Regnase1 is also expressed in lymphoid cells (and appears to be higher than myeloid cells) (Supplemental Figure 1a), the lower levels seen could be just due to the different cellular composition.

Our response to Comment 3-2:

In addition to the comparison of Regnase-1 protein in AML cells and PMBCs, we confirmed that human HSCs express more Regnase-1 protein than PMBCs (lines 8-11, page 17, Figure 5j). However, we cannot exclude the possibility that the decrease in Regnase-1 protein expression in AML cell lines is due to the differentiation stage of the myeloid cells, and we were unable to estimate the proportion of lymphoid cells in PMBCs. Therefore, we deleted sentences from the Discussion which referred to a causal relationship between Regnase-1 and AML that was based on the analysis of Regnase-1 protein expression levels in AML cell lines.

Minor issues:

1. In Fig. 1c, which one is CD34-HSC?

Our response

Thank you for spotting our mistake. The notation of Figure 1c was an error and we have corrected this in the revised manuscript.

2. In Fig. 2d, are all cells CD34⁻? Is the expansion of CD48⁺LSK shown in Fig 2d related to changes cell cycle?

Our response

The graph in Figure 2d did not show data from CD34⁻ cells. Therefore, we performed an analysis of SLAM using CD34⁻ KSL cells. While CD48⁻ CD150⁺ cells were the main population in CD34⁻ KSL cells from control mice, we observed an increase in the number of CD48⁺ cells in both CD150⁺ and CD150⁻ populations in CD34⁻ KSL cells of Rregnase-1-deficient mice (Supplementary Figure 1b).

Analysis of the cell cycle status of CD48⁺ LSK showed that there are many proliferating cells at G1 and G2/M phases consistent with increased numbers of cells in Regnase-1-deficient mice relative to wild-type mice (Supplementary Figure 1c). We have added this information and explanation to the Results section (lines 3-5, page 9).

3. Fig. 2g show increased % of CD34⁻ cells in donor derived LSK for Regnase-1KO but there is no mention about absolute cell number. Given that the % of donor derived LSK is greatly reduced, this information alone is not sufficient to conclude that self-renewal capacity of CD34⁻HSC is increased (page 10). Also, can the authors explain the difference here (almost no CD34⁺HSC cells vs. increased CD34⁺HSC cells in Fig 1i) and also no difference in proliferation in Fig. 2e?

Our response

In accordance with your suggestion, we confirmed increased CD34⁻HSC numbers in Regnase-1-deficient BM and replaced the graph showing the absolute cell number (Figure 2g).

It is difficult to explain the difference in growth of HSC in Figure 1i and Figure 2e, but we are considering the possibility that reduction of colony-forming ability is caused by unbalanced HSC differentiation and proliferation, for example, by decreased stemness due to excessive cell cycling. However, as this is speculative, we have not mentioned this possibility in the paper. If you recommend that we include this in the Discussion, we would be glad to do so.

4. Fig.3 e,f show significant increase of all cell cycle phases (G0, G1, S/G2M) in Reg1KO CD34⁻HSC but the text mention reduction of G0 phase (page 12, line 3)? Which is correct? If all phases of cell cycle are increased, how to interpret this?

Our response

As you point out, the data in Figure 3e,f indicated a significant increase in the number of CD34⁻ HSCs from Regnase-1 deficient mice at all phases of the cell cycle, including proliferating cells in G1 and G2/M and quiescent cells in

G0. However, the proportion of G1 and G2 / M phase cells increased and G0 phase cells decreased. We have corrected this in the revised manuscript (lines 9-12, page 12).

5. Page 21, line 13 in discussion, should be “promoting” cell cycle progression, not suppressing.

Our response

Thank you for spotting this mistake. We have corrected this in the revised manuscript (lines 12, page 21).

6. Page 21, Line 16-17, please clarify where “Regnase-1 is down-regulated in human AML cells and that its expression is inversely associated with AML prognosis” is shown.

Our response

Related to Major issues 3, Regnase-1 protein expression in AML cell lines is lower than in PBMC, but this may be due to the differentiation status of the hematopoietic cells. To avoid ambiguity, we have now omitted this sentence.

7. Page 25 line 13, not sufficient evidence to suggest that Regnase-1 is the “causative gene for human AML. Line 16, “Regnase-1 might be useful for predicting AML prognosis” contradicts the author the main text that Ragnase 1 expression is not associated with prognosis.

Our response

We deleted these parts because our data are insufficient to conclude that Regnase-1 is the causative gene of AML, and mRNA expression of Regnase-1 is not related to prognosis.

Reviewers' comments:

Reviewer #2 (Remarks to the Author):

Kidoya et al.

The authors have improved the previous version of the paper, and many of our "issues" have been taken care of. Concerning the data referred to in Fig. 6, many of the effects they attribute to Regnase-1 could be due to overexpression of the protein in their various experimental systems. However, a potentially very important finding is the study illustrated in Fig. 6d, in which they demonstrate decreased decay of endogenous Gata2 and Tal1 mRNAs in Regnase-1 deficient CD34- HSC, whereas CD34 mRNA was unaffected. Since no overexpression was involved in this experiment, this could be a powerful demonstration of a direct effect on mRNA stability of Regnase-1.

However, there are several aspects of this experiment that deserve further description. For example, what were the starting levels of Gata2 and Tal1 mRNAs in this experiment before the Act D addition? When they say in the figure legend for Fig. 6C that (n = 3 per group), do they mean that cells from three mice were used in each group, or only replicate cultures from one mouse? They refer to one of their earlier papers for real-time RT-PCR methods, but it would be useful to know how they did normalization in their real-time RT-PCR. Finally, I couldn't find (but may have missed) any description of putative binding sequences in the two potential target mRNAs, or any in vitro evidence of direct binding to these targets. All of these pieces of information would strengthen the authors' conclusion that these mRNAs are direct Regnase-1 targets.

Reviewer #3 (Remarks to the Author):

Many concerns still remain in the revised manuscript:

1. Page 10, Line 10-11 "self-renewal capacity of Reg1 Δ/Δ CD34- HSCs was increased" -does not have appropriate data to support this claim. Serial-transplantation assay is required to determine self-renewal capacity.
2. Page 11, line 2-3 "HSC exhaustion may be responsible..." without measuring HSC self-renewal in serial-transplantation, it is difficult to reach this conclusion.
3. If Reg1 Δ/Δ HSCs have increased cell cycling as the authors show (Fig 3e,f), there should be increased apoptosis after 5FU, which is a cell cycle dependent chemotherapeutic agent. However, the authors show increased HSC after 5FU treatment. How are these results reconciled?
4. The new section subtitle "Regnase-1 is necessary for normal hematopoiesis" to describe the "illness" is very vague; it does not provide more clarity to the observations. Are there increase or change in blood counts for the ill mice? This is critical information for assessment of pathology particularly leukemia.
5. Page 16, line 8-11: what is the AML-like phenotype if there were no blasts? It seems that abnormal differentiation is transferred to the recipient mice but they do not develop "illness". If this was the case, these results do not support "AML-like" diagnosis.
6. Figure 5C Table is very confusing. What is the definition of "normal blast" (left column)? How was "normal" vs. "abnormal" populations (blasts, neutrophil, monocyte, lymphocyte) distinguished?
7. The gene expression analysis/comparison with human patients is difficult to follow. It appears that only 1 patient for each subtype (M1, M2, M3) was used. How is the gene expression similarity between the different subtype of patient? It is not clear how significance can be obtained with one patient from each subtype.

8. Figure 7c-h results are interesting; however, there does not appear to be a scrambled or non-silencing shRNA control. This is a critical control to ensure specificity of observed changes.

9. The comparison of Regnase-1 protein levels in AML cell lines and HSC has little relevance in the context of the paper as written; there is no information regarding any association with GATA2 and TAL1 levels in AML cells.

Response to Reviewers' comments:

Reviewer #2 (Remarks to the Author):

Kidoya et al.

The authors have improved the previous version of the paper, and many of our “issues” have been taken care of. Concerning the data referred to in Fig. 6, many of the effects they attribute to Regnase-1 could be due to overexpression of the protein in their various experimental systems. However, a potentially very important finding is the study illustrated in Fig. 6d, in which they demonstrate decreased decay of endogenous Gata2 and Tal1 mRNAs in Regnase-1 deficient CD34- HSC, whereas CD34 mRNA was unaffected. Since no overexpression was involved in this experiment, this could be a powerful demonstration of a direct effect on mRNA stability of Regnase-1.

However, there are several aspects of this experiment that deserve further description. For example, what were the starting levels of Gata2 and Tal1 mRNAs in this experiment before the Act D addition? When they say in the figure legend for Fig. 6C that (n = 3 per group), do they mean that cells from three mice were used in each group, or only replicate cultures from one mouse? They refer to one of their earlier papers for real-time RT-PCR methods, but it would be useful to know how they did normalization in their real-time RT-PCR. Finally, I couldn't find (but may have missed) any description of putative binding sequences in the two potential target mRNAs, or any in vitro evidence of direct binding to these targets. All of these pieces of information would strengthen the authors' conclusion that these mRNAs are direct Regnase-1 targets.

Our response to this Comment:

Regarding the experiment shown in Figure 6d, mRNA expression level at “0h” referred to cells that had been cultured for 1 hour after sorting and collected immediately after adding ActD. Therefore, it seems that the expression level of Gata2 and Tal1 mRNA is almost equivalent to the time before Act D addition. We have now described the procedure for this experiment in the Materials and Methods section. This experiment was performed by using cells from three mice independently in each group. Data are shown as fold-change relative to 0hr treatment, and we performed this analysis referring to previous reports (Uehata T, et al. Cell, 2013). We have added this information and explanation to the M&M section and Figure legend (lines 11-15, page 33 and line 18, page 50).

We contend that in vitro evidence for direct binding of Regnase-1 to the 3'UTR region of Tal1 and Gata2 is provided by the luciferase reporter assay in Figure 6a. In this experiment, a plasmid in which the 3'UTR regions of Gata2 and Tal1 mRNA are linked to luciferase cDNA and a Regnase-1 expression plasmid was simultaneously transfected into the cells. Degradation of Luciferase mRNA by its RNase activity is induced only when Regnase-1 binds to the target 3'UTR sequence, and results in decreased luciferase activity. We understand the reviewer's concern, but this experiment was performed by referring to the previous reports showing Regnase-1 activity (Matsushita K, et al. Nature. 2009). We are hopeful of the reviewer's understanding.

Reviewer #3 (Remarks to the Author):

Many concerns still remain in the revised manuscript:

Comment 3-1:

1. Page 10, Line 10-11 “self-renewal capacity of Reg1 Δ/Δ CD34⁻ HSCs was increased” -does not have appropriate data to support this claim. Serial-transplantation assay is required to determine self-renewal capacity.

Our response to Comment 3-1:

As the reviewer points out, self-renewal capacity of Reg1 Δ/Δ CD34⁻ HSCs cannot be formally shown by our experimental results. Therefore, we removed the word of “self-renewal capacity” and corrected the sentence (lines 10-11, page 10).

Comment 3-2:

2. Page 11, line 2-3 “HSC exhaustion may be responsible...” without measuring HSC self-renewal in serial-transplantation, it is difficult to reach this conclusion.

Our response to Comment 3-2:

We have deleted these phrases because we agree that our data are insufficient to unequivocally conclude that Regnase-1 deletion causes HSC exhaustion. We described this faithfully in the experimental results as “Regnase-1 deficiency results in abnormal regulation of cell growth and induces excessive proliferation” (lines 2-3, page 11).

Comment 3-3:

3. If Reg1 Δ/Δ HSCs have increased cell cycling as the authors show (Fig 3e,f), there should be increased apoptosis after 5FU, which is a cell cycle dependent chemotherapeutic agent. However, the authors show increased HSC after 5FU treatment. How are these results reconciled?

Our response to Comment 3-3:

As indicated by the reviewer, it is expected that apoptosis of proliferating cells would be induced by 5-FU treatment, and actually we did observe decreases of LSK-HSC in Regnase-1-deficient mice 2-4 days after 5-FU treatment. However, there are many quiescent G0 phase cells in LSK-HSC of Regnase-1-deficient mice (Fig 3e,f), and the rapid proliferation of these cells may lead to increases of LSK-HSC 6 days after 5-FU treatment. If it is required, we will add this as a discussion point.

Comment 3-4:

4. The new section subtitle “Regnase-1 is necessary for normal hematopoiesis” to describe the “illness” is very vague; it does not provide more clarity to the observations. Are there increase or change in blood counts for the ill mice? This is critical information for assessment of pathology particularly leukemia.

Our response to Comment 3-4:

As suggested by the reviewer, although Regnase-1-deficient mice had increased white blood cells and blasts in the peripheral blood, blasts were not found in secondary transplanted mice and the phenotype is not consistent with leukemic pathology. Therefore, “illness” is not an appropriate expression for indicating the phenotype of Regnase-1-deficient mice. We altered this to “abnormal hematopoiesis”, and corrected the section subtitle to “Regnase-1 is necessary for hematopoiesis” (lines 1-2, page 14 and line 16, page 13).

Comment 3-5:

5. Page 16, line 8-11: what is the AML-like phenotype if there were no blasts? It seems that abnormal differentiation is transferred to the recipient mice but they do not develop “illness”. If this was the case, these results do not support “AML-like” diagnosis.

Our response to Comment 3-5:

Thank you for these important suggestions. We agree that these results of peripheral blood analyses of mice transplanted with Reg1^{ΔΔ} BM cells may not to be sufficient to conclude “AML-like”, as you have indicated. Therefore, we have now altered this throughout the paper by describing that Regnase-1 deficiency results in “abnormal hematopoiesis” instead of an “AML-like” phenotype.

Comment 3-6:

6. Figure 5C Table is very confusing. What is the definition of “normal blast” (left column)? How was “normal” vs. “abnormal” populations (blasts, neutrophil, monocyte, lymphocyte) distinguished?

Our response to Comment 3-6:

In accordance with the reviewer’s suggestion, we corrected Figure 5c by separately showing the blast frequency in another table. Neutrophil, monocyte and lymphocyte abnormalities were judged by a pathologist based on morphological abnormality using Giemsa stained samples (lines 8-16, page 15).

Comment 3-7:

7. The gene expression analysis/comparison with human patients is difficult to follow. It appears that only 1 patient for each subtype (M1, M2, M3) was used. How is the gene expression similarity between the different subtype of patient? It is not clear how significance can be obtained with one patient from each subtype.

Our response to Comment 3-7:

Regarding the GSEA data in Figure 6h, this does not compare a gene set characterizing a subtype from one patient. Here we compare gene groups changed in Regnase-1-deficient mice with human patient BM samples as follows: Based on the GSE 12662 data, we identified genes with high or low expression in AML (18 M1, 19 M2, and 14 M3 patients) relative to normal bone marrow cells (n=5) by the following method: (a) up-regulated genes with a case mean/control mean ratio >2 , t-test p-value $<1e-3$, and case mean expression ≥ 75 th percentile of mean expression; (b) down-regulated genes with a case mean/control mean ratio $<1/2$, t-test p-value $<1e-3$, and control mean expression ≥ 75 th percentile of mean expression. Gene expression data of Regnase-1-deficient mice were correlated with orthologous human genes, and whether the AML signature genes were coordinately expressed in LT-HSCs from Regnase-1 deficient mice was assessed by GSEA. We then estimated by statistical methods whether gene expression alteration by Regnase-1 deletion in mice HSC is similar to human AML. We have added this information and explanation to the M&M section (line 13, page 37 – line 7, page 38).

Comment 3-8:

8. Figure 7c-h results are interesting; however, there does not appear to be a scrambled or non-silencing shRNA control. This is a critical control to ensure specificity of observed changes.

Our response to Comment 3-8:

Regarding the knockdown experiments shown in Figure 7, we used scrambled shRNA as a control for all experiments. We have corrected Figure 7 and added this information to the M&M section and Figure legend (line 18, page 34 – line 2, page 35, page 51).

Comment 3-9:

9. The comparison of Regnase-1 protein levels in AML cell lines and HSC has little relevance in the context of the paper as written; there is no information regarding any association with GATA2 and TAL1 levels in AML cells.

Our response to Comment 3-9:

We agree with the reviewer's opinion. In accordance with this suggestion, we have deleted data on Regnase-1 protein levels in AML cell lines and HSCs.

REVIEWERS' COMMENTS:

Reviewer #3 (Remarks to the Author):

Figure 3. (a-b), (e-f) cell cycle analysis, the authors shows the cell numbers in each cell cycle phase which were all increased in Reg1 deleted HSC including G0 quiescent cells. This is because the number of HSC is dramatically increased in Reg1 deleted mice. What is the frequency of G0 cells in HSC populations? The authors should show this to clarify whether there exist a shift in cell cycle distribution. The response to comment 3-3 in previous review was that there was increased apoptosis in LSK at day 2-4. The authors did not indicate the time point for data presented in (h-i). If the authors had data for day 2-4, and 6, including these data could enhance the clarity of results presented here. The dynamics after 5FU and interpretation deserve some discussion.

The authors removed the claim of AML-like disease and merely state "abnormal hematopoiesis". "Abnormal hematopoiesis" is used to describe any alternations to hematopoietic differentiation/proliferation and is not a "diagnosis" as the authors stated in the text. If the phenotype is not consistent with AML, does it resemble MPN? The authors also changed the section title to "Regnase-1 is necessary for hematopoiesis" but this is not an appropriate subtitle for what is shown.

Response to Reviewers' comments:

Reviewer #3 (Remarks to the Author):

Comment 3-1:

Figure 3. (a-b), (e-f) cell cycle analysis, the authors shows the cell numbers in each cell cycle phase which were all increased in Reg1 deleted HSC including G0 quiescent cells. This is because the number of HSC is dramatically increased in Reg1 deleted mice. What is the frequency of G0 cells in HSC populations? The authors should show this to clarify whether there exist a shift in cell cycle distribution.

Our response to Comment 3-1:

In accordance with your suggestion, we have added figure of frequency of G0 cells in HSC populations in Figure 3f and showed increment of frequency of G0 cells in Regnase-1 deficient mice.

Comment 3-2:

The response to comment 3-3 in previous review was that there was increased apoptosis in LSK at day 2-4. The authors did not indicate the time point for data presented in (h-i). If the authors had data for day 2-4, and 6, including these data could enhance the clarity of results presented here. The dynamics after 5FU and interpretation deserve some discussion.

Reference

Comment 3-3 in previous review:

3. If Reg1A/Δ HSCs have increased cell cycling as the authors show (Fig 3e,f), there should be increased apoptosis after 5FU, which is a cell cycle dependent chemotherapeutic agent. However, the authors show increased HSC after 5FU treatment. How are these results reconciled?

Our response to Comment 3-3 in previous review:

As indicated by the reviewer, it is expected that apoptosis of proliferating cells would be induced by 5-FU treatment, and actually we did observe decreases of LSK-HSC in Regnase-1-deficient mice 2-4 days after 5-FU treatment. However, there are many quiescent G0 phase cells in LSK-HSC of Regnase-1-deficient mice (Fig 3e,f), and the rapid proliferation of these cells may lead to increases of LSK-HSC 6 days after 5-FU treatment. If it is required, we will add this as a discussion point.

Our response to Comment 3-2:

Reviewer may have confused apoptosis analysis data (Fig3 h,i) with 5-FU analysis data (Fig3 k,l). Fig3 h,I show the result of apoptosis analysis of HSCs from adult mice without any experimental treatment and not from 5-FU treated mice. In the previous response to comment 3-3, we described the possibility that apoptosis of HSCs is involved in the response to 5-FU. In accordance with reviewer's suggestion, we have added this consideration to

manuscript (lines 10-12, page 13).

Comment 3-3:

The authors removed the claim of AML-like disease and merely state “abnormal hematopoiesis”. Abnormal hematopoiesis” is used to describe any alternations to hematopoietic differentiation/proliferation and is not a "diagnosis" as the authors stated in the text. If the phenotype is not consistent with AML, does it resemble MPN?

Our response to Comment 3-3:

Thank you for these important suggestions. As you pointed out, similarity with MPN is observed such as increase of abnormal granulocytes in the peripheral blood and bone marrow of Regnase-1 deficient mice. We have added this information to manuscript (lines 18, page 15).

Comment 3-4:

The authors also changed the section title to “Regnase-1 is necessary for hematopoiesis” but this is not an appropriate subtitle for what is shown.

Our response to Comment 3-4:

We agree with the reviewer’s opinion. In accordance with this suggestion, we have corrected the section title to “Regnase-1 deficiency causes impaired hematopoietic function”.